# Expression Profiles of Housekeeping Genes and Tissue-Specific Genes in Different Tissues of Chinese Sturgeon (*Acipenser sinensis*)

**DOI:** 10.3390/ani14233357

**Published:** 2024-11-21

**Authors:** Yanping Li, Yunyun Lv, Peilin Cheng, Ying Jiang, Cao Deng, Yongming Wang, Zhengyong Wen, Jiang Xie, Jieming Chen, Qiong Shi, Hao Du

**Affiliations:** 1Key Laboratory of Freshwater Biodiversity Conservation, Ministry of Agriculture and Rural Affairs of China, Yangtze River Fisheries Research Institute, Chinese Academy of Fishery Sciences, Wuhan 430223, China; liyanping_sci@foxmail.com (Y.L.); lvyunyun_sci@foxmail.com (Y.L.); chengpeilin@yfi.ac.cn (P.C.); 2Key Laboratory of Sichuan Province for Fishes Conservation and Utilization in the Upper Reaches of the Yangtze River, College of Life Science, Neijiang Normal University, Neijiang 641100, China; jiangying29@126.com (Y.J.); wym8188@126.com (Y.W.); zhengyong_wen@126.com (Z.W.); xiejianggz@163.com (J.X.); 3Department of Bioinformatics, DNA Stories Bioinformatics Center, Chengdu 610000, China; dengcao@dnastories.com; 4Shenzhen Key Lab of Marine Genomics, BGI Academy of Marine Sciences, BGI Marine, Shenzhen 518081, China; chenjieming@chinamarine.com; 5Laboratory of Aquatic Genomics, College of Life Sciences and Oceanography, Shenzhen University, Shenzhen 518057, China

**Keywords:** Chinese sturgeon, transcriptome sequencing (RNA-seq), housekeeping gene, tissue-specific gene, co-expression, ribosomal protein gene

## Abstract

The Chinese sturgeon (*Acipenser sinensis*), an ancient fish with important ecological and evolutionary significance, has been listed as a threatened species. Comprehensive gene expression profiles across different tissues provide essential information for understanding their biological functions in Chinese sturgeon. Our current study involves a thorough analysis of gene expression to identify and characterize housekeeping genes and tissue-specific genes, as well as potential hub genes across eleven tissues. These findings provide a fresh perspective on transcriptomic regulation in Chinese sturgeon.

## 1. Introduction

Chinese sturgeon (*Acipenser sinensis*) is a large, typical anadromous species. Its natural geographical distribution area is the East Asian continental shelf, and its main inflow river system includes the Yangtze River, Yellow River, Qiantang River, Minjiang River, and Xijiang River of the Pearl River Basin [1]. Due to overfishing, habitat destruction and reduction of spawning grounds, water pollution, and various anthropogenic activities, the wild Chinese sturgeon populations have experienced a dramatic decline in the past thirty years, with a sharp reduction in distribution areas. At present, Chinese sturgeon only appears in the Yangtze River, with the only known spawning ground located within 4 km downstream of the Gezhouba Dam; no traces of any Chinese sturgeon have been found in other rivers as its natural distribution regions [1,2]. The Chinese sturgeon is classified as a national first-class priority for national aquatic animal conservation, and it has been designated as an extremely critical species by the International Union for Conservation of Nature (IUCN) [3], as well as continuously recognized as a threatened species on the IUCN Red List in 2022 (https://www.iucnredlist.org/species/236/219152605 (accessed on 2 November 2024)). It is also listed as a protected species under Appendix II of the International Convention on Trade in Endangered Species of Wild Fauna and Flora [4].

The natural breeding of Chinese sturgeon was disrupted for three consecutive years (2017–2019), resulting in a critical decline in the natural population and leading it to the brink of extinction [2]. The Gezhouba Dam across the Yangtze River has been identified as the primary factor contributing to the decline in the natural population of Chinese sturgeon [5]. This situation has considerable implications for habitat fragmentation, obstruction of migration pathways, and disruption of spawning habitats, which ultimately lead to a decline in biodiversity. This specific issue can be attributed to the construction of Gezhouba Dam, which has shortened the migratory distance, reduced the survival rate, and delayed the breeding season of the wild Chinese sturgeon populations. This situation is believed to be the main reason for the decline of wild Chinese sturgeon stocks. In addition, the water storage operations associated with large-scale cascade hydropower projects in the upper reaches of the Yangtze River, notably the Three Gorges Dam, have been identified as a major factor disrupting the natural reproduction of Chinese sturgeon [6]. In general, various anthropogenic activities, river erosion, channel dredging, sand dredging, flood control measures, and urban development projects have also played a role in diminishing the habitats available for Chinese sturgeon larvae and juveniles. As a result, these factors have led to a decrease in supplementary populations and a continuous reduction in breeding populations of Chinese sturgeon.

Chinese sturgeon, characterized by its large body size, extended lifespan, and extensive migratory patterns, is unequivocally regarded as a flagship and umbrella species within the Yangtze River ecosystem. Thus, the effective conservation of the Chinese sturgeon is crucial for fostering a harmonious relationship between humans and the natural environment, as well as for advancing the sustainable socioeconomic development of the Yangtze River Basin [2]. Moreover, due to its extensive evolutionary history and critical phylogenetic position, the Chinese sturgeon serves as a significant comparative model for the investigations of vertebrate evolution, physiology, and immunology [7,8]. Several research initiatives and conservation programs have focused on artificial breeding and reintroduction of this threatened species in order to facilitate its recovery. Notable multi-omics efforts, including whole-genome sequencing of Chinese sturgeon and its application [9,10], and transcript annotation of Chinese sturgeon using Pacific Biosciences (PacBio) isoform sequencing (Iso-seq) and RNA sequencing (RNA-seq) [11], have significantly increased our understanding of genetic mechanisms of endangerment of the Chinese sturgeon. The availability of multi-omics studies and data focusing on Chinese sturgeon remains limited, primarily due to its polyploid lineages and highly intricate genomic structure. Given the spatiotemporal specificity of gene expression, it is necessary to carry out transcriptomics of Chinese sturgeon in different tissues and ages.

Tissue-specific genes (TSGs) are characterized by markedly elevated expression levels in a particular tissue compared to baseline expression levels in other tissues. Their expression profiling is essential for understanding the development, complexity, and evolutionary history of an organism at a systemic level. Additionally, the categorization of genes based on their expression patterns across various tissues is crucial to gain insights into the molecular mechanisms underlying tissue activity and function. This classification aids in identifying critical regulatory elements and elucidating the relationships between phenotypic traits and the functional evolution of tissues [12,13,14]. Tissue-specific expression profiles have been characterized in various aquaculture species, such as in Atlantic salmon (*Salmo salar*) [15], crucian carp (*Carassius carassius*) [16], lake sturgeon (*Acipenser fulvescens*) [17], rainbow trout (*Oncorhynchus mykiss*) [18], and zebrafish (*Danio rerio*) [19]. In particular, in the case of Atlantic salmon, the identification of some genes that are specifically expressed in the brain–pituitary–gonad axis has contributed to a basic understanding of essential reproduction and life history characteristics in salmonid species [15]. Additionally, in lake sturgeon, for example, an analysis of mRNA transcription across various tissues and within the gastrointestinal tract provided a novel insight into the potential regulatory mechanism of the microbiome in this ancient fish [17].

Housekeeping genes (HKGs) are those genes whose encoded products play a crucial role in fundamental cellular maintenance and biological functions. It has been established that certain genes show variable expression levels that are context-dependent, influenced by factors such as developmental stage, tissue type, and the presence of normal or pathological conditions [20]. In contrast, other genes are transcribed at a relatively constant level across various tissue types and conditions. Thus, a housekeeping function has been inferred or even empirically validated for the majority of these genes, which accounts for both their broad expression patterns and their consistent transcript abundance. The study of HKGs, particularly their significance in determining the fundamental functions of living cells, deserves focused attention due to their potential use as reference points for the experimental evaluation of gene expression. For example, this is illustrated by the identification of an ideal HKG characterized by its constant and potentially increased expression across various tissue types under diverse conditions, which would serve as a valuable resource for normalization and interpretation of expression level data of other studied genes within the biological context [21].

Transcriptomic resources in aquaculture establish a robust foundation for the development of tools for the analysis of the expression of TSGs and HKGs. Subsequent analyses may use tissue-specific expression and co-expression patterns to identify markers within various tissues that could reveal functional characteristics of genes involved in various biological and cellular processes since those genes with analogous functions are likely to show co-expression patterns across different cell types or tissues [22]. To the best of our knowledge, to date, there has been a lack of research focused on the comprehensive characterization of TSGs, HKGs, and their co-expression modules and networks in multiple tissues of Chinese sturgeon. This transcriptomic study will contribute to improving our understanding of the adaptation, development, growth, and metabolic parameters of Chinese sturgeon. The main aim of this study is, therefore, to address the existing gap by providing a detailed catalog of TSGs and HKGs that are expressed in Chinese sturgeon, using advanced multi-tissue deep RNA-seq analysis. Spatial transcriptome, particularly in the context of multi-tissue studies, facilitates the study of the adaptive evolution of genes and establishes a basis for co-expression network analysis across various tissues.

Overall, the aims of this study were first to construct genome-wide gene expression atlases of TSGs, HKGs, and their functional co-expression networks for Chinese sturgeon and second to focus on conducting a comprehensive analysis to identify optimal housekeeping hub genes in Chinese sturgeon, which could serve as reference genes throughout the critical stages of embryonic development and in various differentiated adult tissues. In response to the aforementioned questions, this study provides functionally annotated, high-quality transcriptome assemblies derived from multiple tissues, along with comprehensive genome-wide expression profiles of TSGs, HKGs, and their functional co-expression networks. We also identified ideal HSGs and potential pathways and biological processes associated with housekeeping and co-expression modules in this extremely important aquatic species. The transcriptomic resources generated in this study may prove valuable in elucidating the mechanisms underlying organ biology and functions of specific metabolic pathways across different tissues and provide the foundation to further explore the genetic mechanisms of endangerment for Chinese sturgeon.

## 2. Materials and Methods

### 2.1. Sample Collection and RNA Preparation

A 1-year-old juvenile fish from a Chinese sturgeon conservation base located at Taihu Lake in Jingzhou province of China was anesthetized and dissected to collect different tissues for RNA isolation. Before collection, we conducted a thorough health assessment based on established criteria, including a visual inspection for signs of disease and a behavioral assessment (such as observing swimming patterns). These evaluations were essential in ensuring that the specimen represented a healthy individual within the examined population.

Fish handling and experimental procedures were performed according to Chinese animal protection laws. Eleven tissues, namely the epidermis, notochord cartilage, swim bladder, brain, heart, pronephros, kidney, spleen, gallbladder, liver, and muscle, were sampled on ice, transferred into the centrifuge tube with RNAlater, and promptly stored in liquid nitrogen to preserve RNA integrity. Gloves and face masks were worn at all times, and materials were handled using RNase-free. Our samples do not overlap with those in the earlier research except for the swim bladder and liver [11]. Additionally, the age of the specimens collected in our present study differs from those in the previous report [11]. Total RNA from each sample was individually isolated using TRIzol reagent (Invitrogen, Carlsbad, CA, USA) following the manufacturer’s guidelines. Before constructing library, the quality and integrity of the total RNA were evaluated by a set of quality control procedures. RNA degradation and potential contamination were examined by 1% agarose gel electrophoresis. RNA purity was assessed by measuring the OD260/280 ratio using a Nanodrop 2000 spectrometer (Thermo Fisher Scientific Inc., Waltham, MA, USA). RNA concentration was quantified using the Qubit RNA Assay Kit on a Qubit 2.0 Fluorometer (Thermo Fisher Scientific Inc.), and the RNA integrity number was determined using an RNA Nano 6000 Assay Kit on an Agilent 2100 Bioanalyzer (Agilent Technologies Inc., Santa Clara, CA, USA). After confirming the isolated RNA met high-quality standards, it was used to prepare library and for subsequent applications.

### 2.2. Library Preparation and Sequencing

#### 2.2.1. Preparation and Sequencing of PacBio Single-Molecule Real-Time (SMRT) Sequencing Bell Library

After extraction of total RNA from the eleven tissues, polyadenylated mRNA (poly(A) mRNA) was purified by enrichment with oligo (dT) magnetic beads and used to build the full-length Chinese sturgeon transcriptome library. The library was carefully constructed, following the Iso-seq protocol, using the SMARTer PCR cDNA Synthesis Kit (Takara Bio Inc., Kusatsu, Japan) and the BluePippin Size Selection System (Pacific Biosciences Inc., Menlo Park, CA, USA) protocol, according to the manufacturers’ instructions. Briefly, poly(A) mRNA was subjected to reverse transcription using the SMARTer PCR cDNA Synthesis Kit to generate complementary DNA (cDNA), which was amplified by polymerase chain reaction (PCR) following the BluePippin size-selection system protocol. The library was prepared by fixing the damage to the full-length cDNA, repairing the terminal, and attaching the SMRT dumbbell adapters. Exonuclease digestion was conducted to remove those sequences of the unattached adapters at both ends of the cDNA. The resulting cDNA was then combined with primers and DNA polymerase to construct a complete SMRT bell library. Work surfaces were treated with RNA-decontamination solutions, and laminar flow hoods were utilized during library preparation to provide a sterile environment. After the library was qualified, a PacBio Sequel II platform (Pacific Biosciences Inc.) was applied for sequencing, following the effective concentration of the library and data output requirements. In fact, SMRT sequencing allowed for the real-time observation of DNA synthesis by capturing the activity of DNA polymerases on single DNA molecules. The SMRT sequencing produced long-read sequences, enabling more accurate assembly of the genomes and improved resolution of the complex genomic regions. This technology is particularly beneficial for studying structural variants and for de novo genome assembly.

#### 2.2.2. Library Construction and MGI Sequencing

After confirming the quality of the total RNA samples, poly(A) mRNA was purified by enrichment with magnetic beads coated with oligo(dT) and used to synthesize the corresponding cDNA. The first-strand cDNA was synthesized from poly(A) mRNA using a random hexamer primer, and M-MuLV reverse transcriptase, followed by the degradation of RNA with RNaseH to achieve comprehensive coverage of the transcriptomes, which enabled us to amplify a wide variety of transcripts, including those expressed at lower levels. Subsequently, second-strand cDNA synthesis was achieved using DNA Polymerase I and deoxynucleotide triphosphates.

The purified cDNA underwent terminal repair, followed by the addition of a poly(A) tail and ligation of the sequencing adapter. The double-stranded cDNAs were subjected to heat denaturation and circularization by the splint oligo sequence to generate a single-stranded circular DNA library. After the library was constructed, it was initially quantified with a Qubit RNA Assay Kit on a Qubit 2.0 Fluorometer, and then the insert size of the library was determined with an Agilent 2100 Bioanalyzer. Once the insert size was validated to be about 350 bp, quantitative PCR was performed to accurately quantify the effective concentration of the library. After the library test was qualified, the circular DNA was subjected to rolling circle replication to produce DNA Nano Balls (DNBs), which were loaded into the sequencing chip and further processed using an MGI DNBSEQ-T7 Sequencing Platform (MGI Tech Co., Ltd., Shenzhen, China).

### 2.3. Full-Length Transcriptome Data Analysis

SMRT analysis software IsoSeq v3.0 [23] was employed to process the PacBio sequencing raw reads and produce high-quality consensus sequences. The circular consensus sequences (CCS) were derived from subread BAM files using the following parameters: min_length 200, min_pass 1, max_length 100,000, and min_read quality 0.9. Because the 3′ primer, 5′ primer, and PolyA were in the target sequences, the CCSs were divided into full-length sequences (FL, full length) and non-full-length (NFL, non-full length) sequences. The primer and poly(A) tail sequences were excised from both ends of the CCS, followed by elimination of the artificial concatemers within the full-length sequence to generate the full-length non-chimeric sequence (FLNC) and NFL. Additionally, the strand orientation of the sequence was determined by leveraging the primer and poly(A) tail data. The FLNC reads were clustered using IsoSeq v3.0 to obtain non-redundant, high-quality, full-length transcripts. Finally, the cd-hit-est v4.8.1 software [24] was applied to cluster the FLNC sequences, considering transcripts within the same cluster as different splice variants of the same gene to obtain the consensus isoforms derived from the clustering, thereby selecting high-quality sequences for further analysis.

*A*. *sinensis* samples were sequenced on a DNBSEQ-T7 platform (MGI Tech Co., Ltd., Shenzhen, China) at Frasergen Bioinformatics Co. Ltd. (Wuhan, China). The FASTP v0.21.1 program [25] was employed to process the raw data with the options “--detect_adapter_for_pe --trim_front1 5--trim_tail1 5--trim_front2 5--trim_tail2 5--cut_right--cut_window_size 4--cut_mean_quality 20--length_required 50--n_base_ limit 3--qualified_quality_phred 20 --unqualified_percent_limit 10”. In this phase, we obtained clean data by eliminating reads containing adapter sequences, reads with Poly-N sequences, and low-quality reads. Subsequent analyses were performed solely on those high-quality, clean data.

### 2.4. Structural and Functional Annotation of Unigenes

The GenomeThreader v1.7.3 software [26] was applied to annotate structural details in the gene set, opting for the longest as the representative coding sequences (CDS). Subsequently, the TransDecoder v5.5.0 pipeline [27] was employed to annotate structural information that was not annotated by the GenomeThreader, selecting more of the longest CDS annotation outcomes. Ultimately, the two sets of results were merged, and the longest CDS was selected based on the corresponding relationship from cd-hit-est v4.8.1 software [24] as the structural annotation result of the gene locus. Protein sequences of the outcome of structural annotation were enriched with functional annotations by searching public databases, including Gene Ontology (GO) [28], Kyoto Encyclopedia of Genes and Genomes (KEGG) [29], InterProScan (Iprscan), SwissProt (a manually annotated and reviewed protein sequence database) [30], and the National Center for Biotechnology Information (NCBI) non-redundant protein sequences (Nr) [31]. The blastp outcomes from the NCBI NR and SwissProt databases underwent additional refinement to identify the most suitable alignments based on selection criteria, including identity, query coverage, and subject coverage ≥ 30.

### 2.5. Quantifying Transcript Abundance and Correlation Analysis

Gene expression profiles, including gene counts and TPM (Transcripts Per Million) values, were obtained for each sample. Quantification and merging transcript expression of all samples were performed using the align_and_estimate_abundance.pl script in the Trinity v2.14.0 package with the est_method salmon parameter [27]. We chose the “est_method” parameter as “salmon” due to its efficiency in quantifying transcript abundance from RNA-seq data. Salmon utilizes advanced statistical models that allow accurate estimates of expression levels, which was suitable for our dataset. Its ability to perform quasi-mapping enabled us to quantify transcripts efficiently.

Principal component analysis (PCA) was performed using the TPM expression values of those orthologous genes from the eleven *Chinese Sturgeon* tissues. Pearson’s correlation analysis was also conducted among the eleven samples. Related results of the correlation analysis among tissues were obtained and visualized using the PtR (Perl-to-R) script in the Trinity software (version 2.15.2).

### 2.6. Identification and Enrichment of TSGs and HKGs

The products of TSGs, also known as luxury genes, are expressed specifically in different cell types and can lead to development of specific morphological and physiological characteristics in various cells. HKGs, also known as steward genes, are a group of genes that must be continually expressed in all cells, and their products are necessary for maintaining the basic activities of various cells. In short, HSGs are constitutively expressed across all cell types, while TSGs are expressed in specific groups of cells.

A tau (τ) expression index is the most appropriate metric for assessing gene tissue specificity [32]. This index ranges from 0 to 1 and represents transcripts exhibiting high tissue specificity with values close to 1 (τ > 0.85), while transcripts that are broadly expressed, such as HKGs, tend to have a tissue specificity index closer to 0 (τ < 0.3) [12,32,33]. In addition, the counts and Gini coefficient are important indicators for assessing gene expression levels in specific tissues. The higher the values of the three indicators, the more specific the gene expression in the target tissue. In contrast, the lower the values, the more conservative the expression. In this study, tspex v 0.6.1 [34] was applied to compute the three parameters. Default settings were used to calculate τ and the Gini coefficient, while the count threshold was set to one. Subsequently, the outcomes were leveraged for the additional delineation of TSGs and HKGs. In this study, a threshold value of 0.8 was set to identify TSGs with a minimum of two of the three indicators exceeding 0.8. These TSGs and HKGs were validated using the R package UpSetR v4.3.1 [35] to generate a Venn diagram in which circular dots represent the connections between different entries. The presence of one dot indicated that the entry was a TSG. Similarly, a threshold value of 0.2 was set for filtering the HSGs. When a gene had a value < 0.2 for two out of the three indicators, it was considered an HKG if there were connecting dots in each entry in the Venn diagram generated by UpSetR.

In addition, GO functional enrichment and KEGG pathway enrichment of the TSGs and HKSs were performed using custom scripts. GO terms and KEGG pathways with a false discovery rate (FDR) ≤ 0.05 were considered significant. Net plots were visualized using the R package aPEAR (Version 1.0.0) [36]. We generated a graph showing the optimal number of clusters for K-means clustering of the HKGs. The K-means algorithm is widely used for clustering analysis based on the Euclidean distance, which considers that the closer the distance between two objects, the greater the similarity. We employed the R packages factoextra v1.0.7 [37] and ggplot2 v3.4.3 [38] to compute metrics, such as elbow coefficient, silhouette coefficient, and Gap statistic, aiding in identification of the optimal number of clusters [39]. Subsequently, the ComplexHeatmap package (Version 2.22.0) [40] was applied to generate a heat map, displaying the K-means clustering numbers. GO term enrichment and KEGG pathway enrichment were also performed on each cluster of obtained HKGs from the best clustering division by k-means.

### 2.7. Co-Expression of the HKGs and Hub Genes and Network Module Analysis

We conducted a weighted gene co-expression network analysis (WGCNA) based on the HKGs to identify gene modules with coordinated expression of HKGs and to investigate the relationship between the gene networks and specific tissues of interest. Before conducting this analysis, we refined the chosen gene set by eliminating low-quality genes or samples that could potentially introduce instability in the results so as to enhance the precision of those predicted networks. WGCNA was performed using the WGCNA R package v. 1.70.3 following a previously published procedure [41]. The data input for the WGCNA was the TPM expression matrix of the HKGs after a log2 transformation.

First, co-expression similarity was determined based on the expression values using Pearson’s correlation coefficient. Correlation coefficients of the expression of these genes were used as a soft threshold based on a power function and scale-free topology model fit. An adjacency matrix was calculated by exponentiation of the correlation matrix of gene expression profiles to the selected power β (soft threshold) to maintain high similarity measures as high adjacencies, while lower similarity measures were adjusted towards an adjacency of zero [42]. The β value was determined according to the scale-free topology criterion [43]. A linear regression model fitting index (R^2^) was used to determine how well the network met this criterion. We chose a β value in the interval (1, 20), which optimized the scale-free topology model fit (R^2^ ≥ 0.8). Subsequently, to identify gene clusters within the dataset, an adjacency matrix was used to calculate topological overlap measure, which quantifies the extent of shared neighbors between gene pairs within the network [44].

All gene models exhibiting similar expression patterns were detected using the gene cluster dendrogram and the dynamic tree-cut algorithm [41,45]. Subsequently, all genes contained in each module were analyzed to determine their association with a given tissue. Those with a strong correlation and a significance level of *p* < 0.05 were considered modules related to the target tissue. The modules defined above were selected based on their module correlation, and the eigengene expression levels were analyzed for the modules using the WGCNA package [41]. Retained genes in the modules were further analyzed by functional enrichment analysis. GO categories and KEGG pathways with an FDR ≤ 0.05 were deemed as significant. However, only those significant terms and pathways associated with a minimum of three genes were retained. If no output file was generated, it indicated that the module had no enrichment results or there were less than two related GO terms available for network visualization.

Hub genes were selected from each key functional module to investigate the structure of the target network. This involved sorting genes within each module based on their internal connectivity in a descending order and selecting the top 5% as the hub genes of that module. The network was visualized using the igraph package in R (Version 2.1.1) [46].

## 3. Results

### 3.1. Summary of the PacBio Iso-Seq Data

In our current study, using PacBio Sequel technology, a total of 806,926 polymerase reads were generated with an average length of 103,422 bp and an N50 length of 193,380 bp. PacBio ISO-seq generated a total of 46,605,450 subreads, from which 318,019 CCSs were obtained by self-correction with an average length of 2200 bp. Among these CCS reads, 108,170 were identified as FLNC reads with an average length of 2166 bp. The FLNC reads were subsequently clustered using isoseq3 to obtain non-redundant, high-quality, full-length transcripts. A total of 25,434 polished high-quality consensus reads were generated, with an average length of 2691 bp and an N50 length of 3195 bp (Table 1).

### 3.2. Function Annotation of Unigenes and Analysis of Transcripts

In order to generate comprehensive information on the gene function of the predicted unigenes, the transcripts were subjected to annotation analysis using five public databases, including Iprscan, GO, KEGG, NR, and SwissProt. Our results revealed that among a total of 25,434 unigenes, 20,887 were successfully annotated with at least one hit. A total of 19,664 and 13,628 transcripts were annotated to the Iprscan and GO databases, respectively, while 11,535, 18,288, and 14,671 transcripts were annotated to the KEGG, NR, and SwissProt databases, respectively (Figure 1A). In total, 8722 transcripts were annotated in all databases (Figure 1B).

To identify biological pathways active in Chinese sturgeon, we used annotated sequences for GO and KEGG pathway enrichment. The GO analysis revealed the enrichment of unigenes mainly to three GO categories, including molecular function (MF), cellular component (CC), and biological process (BP). The KEGG pathway analysis revealed the enrichment of unigenes predominantly in five different pathways, namely cellular processes, environmental information processing, genetic information processing, metabolism, and organismal systems. The bar charts representing GO terms and KEGG pathways were limited to show the top 10 entries in each major category. The terms nucleic acid binding, protein kinase activity, actin filament binding, and transmembrane signaling receptor activity were prominently featured within the molecular function classification. The transcripts classified under the cellular component category showed a notable enrichment in terms related to cytoplasm, extracellular space, endoplasmic reticulum, and plasma membrane. The processes in the biological process category that exhibited the most enrichment were signal transduction, DNA transposition, and G-protein-coupled receptor signaling pathway (Figure 2A). In the KEGG analysis, thermogenesis, NOD-like receptor signaling pathway, oxytocin signaling pathway, and antigen processing and presentation were the most enriched processes in the category of organismal systems. Among the category of metabolism, a notable enrichment was observed in several pathways related to metabolic pathways, carbon metabolism, oxidative phosphorylation, and biosynthesis of amino acids. RNA transport, protein processing in the endoplasmic reticulum, and ribosome biogenesis were the most enriched processes in the category of genetic information processing. The PI3K−Akt signaling pathway, Rap1 signaling pathway, and cell adhesion molecules were the prominently featured processes in the environmental information processing classification. Endocytosis, focal adhesion, cellular senescence, and apoptosis were the most highly represented processes in the cellular processes category (Figure 2B).

### 3.3. Correlation of the Elven Chinese Sturgeon Tissues

We employed two methods to compare the similarity between transcriptomes from different Chinese sturgeon tissues. The first approach involved Pearson’s correlation analysis, which revealed that muscle had a relatively higher correlation with notochord cartilage and heart than other tissues. Similarly, the brain had a higher correlation with notochord cartilage, heart swim bladder, and pronephros, but its Pearson correlation coefficient (r) values with other tissues were all <0.8. The kidney showed the strongest correlation with pronephros, followed by a significant correlation with the heart, gallbladder, liver, and swim bladder. Regarding the epidermis tissue, it showed significant correlations with notochord cartilage, heart, swim bladder, pronephros, and spleen (r > 0.8), but its correlations with other tissues were relatively weak. Conversely, the correlation between gallbladder and liver was found to be the highest among the evaluated tissues (r = 0.99). Additionally, a significant correlation was found between notochord cartilage and swim bladder, as well as between spleen and pronephros, with a Pearson correlation coefficient value of 0.91 (Figure 3A).

The second approach involved a widely utilized PCA ordination method to represent variations among transcriptomes in a multidimensional space, where PC1 and PC2 accounted for 50% of the total variability (Figure 3B), and PC2 and PC3 accounted for 33.33% of the total variability (Figure 3C). For the comparison of inter-tissue relevance, the patterns derived from PCA had similarities to the outcomes obtained by Pearson’s correlation analysis. Notably, a minimal correlation was observed between the brain and muscle compared to other tissues, whereas the most significant correlation was identified between the gallbladder and liver (Figure 3B,C). Additionally, a notable correlation was found among the tissues of the epidermis, notochord cartilage, heart, spleen, swim bladder, and pronephros (Figure 3B), which is consistent with the findings based on the Pearson correlation coefficients.

### 3.4. Quantifying Transcript Abundance and Expression Patterns in the Chinese Sturgeon Transcriptomes

Upon analyzing the spread of transcript abundance across various tissues in different TPM intervals, it was clear that the majority of genes exhibiting low expression levels were observed across the eleven distinct tissues. In the brain, kidney, spleen, and swim bladder tissues, there was a slightly increased proportion of TPM intervals within the 50–100 range compared to those within the 100–500 range. Conversely, in the remaining tissues, the ratios of values within the two ranges had a comparable distribution.

The distribution of tissues with high expression levels of TPM is relatively uniform across the eleven different types of tissues, with the lowest proportion (Appendix A). Additionally, a detailed analytical investigation using a boxplot (Appendix A) and a density distribution plot (Appendix A) provided a visually clear perspective, reinforcing our finding that the brain showed the highest distribution of unigene expression levels compared to other tissues. The levels of unigene expression across various tissues, including swim bladder, kidney, heart, notochord cartilage, epidermis, pronephros, and spleen, exhibited a notable concentration, with slightly decreased levels in the brain and the lowest levels detected in the muscle. The unigene expression levels in the liver and gallbladder tissues showed a modest increase compared to those in the muscle (Appendix A).

### 3.5. Identification and Characterization of TSGs and HKGs for Functional Enrichment

The analysis of TSG expression patterns based on the transcriptomes of eleven different tissues revealed that the number of TSGs ranged from 25 to 2073 across the eleven tissues, with the brain exhibiting the highest and the liver showing the lowest. Notably, the gallbladder and liver shared a similar number of TSGs, with only 27 genes identified in the gallbladder. There were 64 and 70 genes specifically concentrated in the pronephros and notochord cartilage tissues. In addition, the muscle, heart, and swim bladder had 114, 157, and 160 TSGs, respectively. In contrast, the spleen, skin, and kidney had 207, 242, and 307 TSGs, respectively (Figure 4A). To further explore the biological information of these TSGs, GO term and KEGG pathway enrichment analyses were conducted based on all TSGs. The network diagram of GO term enrichment for TSGs was limited to displaying the terms with more than two nodes, while the KEGG-enriched pathways were exclusively found when the *p* ≤ 0.05. Enriched GO terms were mainly involved in voltage-gated potassium channel activity, transporter activity, G-protein-coupled receptor activity, glial cell differentiation, and synaptic vesicle (Figure 4B). Among them, the voltage-gated potassium channel is a type of ion channel on the cellular membrane that is involved in maintaining the equilibrium of potassium ions within and outside the cell. This voltage-gated potassium channel is crucial for various physiological processes, such as nerve conduction, muscle function, and cardiac rhythm regulation.

Enriched KEGG pathways were predominantly associated with the oxytocin signaling pathway, taste transduction pathway, serotonergic synapse pathway, glutamatergic synapse pathway, and the neuroactive ligand–receptor interaction pathway (Figure 4C). Functionally significant terms and pathways based on brain TSGs were mainly related to nervous system development, voltage-gated potassium channel activity, ionotropic glutamate receptor activity, monoatomic ion transport, and neuroactive ligand−receptor interaction, which exhibited a similar pattern to the enrichment results based on genes specific to all the eleven tissues (Appendix A).

The HKG identification results revealed the presence of 787 HKGs with stable expression across all eleven tissues (Figure 5A). In order to delve deeper into the biological characteristics of HKGs, we performed functional enrichment analysis using the GO and KEGG databases. We observed that the terms and pathways showing significant enrichment of HKGs were consistent with those identified data based on the results of TSG enrichment. Functionally significant terms and pathways based on HKGs were primarily associated with elongation factor activity, ribosome, translation initiation factor activity, proteasome core transcription complex, proteolysis involved in protein catabolic process, ribosome binding, and RNA transport (Figure 5B,C). Significantly enriched terms and pathways based on the HKGs were mostly important components and pathways related to protein synthesis and metabolism.

### 3.6. Determining the Optimal Clusters for HKGs and Enrichment of Each Cluster

The coefficient of variation (CV), also referred to as the dispersion coefficient, is a statistical metric for quantifying the extent of variability in individual observation within a dataset. High CV values for a gene indicate its unstable expression levels across various samples. Each gene is assigned a CV value, which is then used to determine its variability. In our present study, CV values for all genes were calculated and categorized into three groups denoting high, moderate, and low variability and determined by their upper and lower quartile thresholds. Those genes with CV values above the upper quartile were categorized as high variability, accounting for 25% of the entire dataset. Those within the interquartile range as moderate variability accounted for 50% of the dataset, and those below the lower quartile as low variability accounted for 25% of the total dataset. HKGs, known for their stable expression levels, were expected to fall into the low variability group (Appendix A). These results shown in Appendix A revealed that HKGs made up a small proportion (2.9%) of the entire dataset. In addition, the expression levels of HKGs were further analyzed based on TPM values, with the categorization of three groups, including those with TPM values above 50, between 10 and 50, and between 1 and 10 (in fact, those genes with TPM values below 1 were excluded).

Three different methods, namely elbow coefficient, silhouette coefficient, and Gap statistic, were employed to determine the optimal number of clusters for the HKGs. This cluster number was determined to be four (Appendix A). Measurement of the gene expression levels of constitutively expressed genes revealed that cluster 1 had the highest level of expression, while cluster 2 had the lowest expression level but with the highest proportion of genes. Cluster 3 contained the fewest genes but had expression levels slightly lower than those of cluster 1. Conversely, cluster 4 had expression levels slightly higher than those of cluster 2 despite having slightly fewer genes than cluster 2 (Appendix A).

Finally, GO term and KEGG pathway enrichment was conducted based on individual gene clusters derived from the optimal clustering division determined by K-means as described above. Our results of the analysis were summarized as follows. Functionally significant terms and pathways of cluster 1 were primarily associated with proteasome core complex, proteolysis involved in protein catabolic process, translation initiation factor activity, longevity regulating pathway, and proteasome (Appendix A). Functionally significant terms and pathways of cluster 2 were mainly related to pseudouridine synthesis, DNA-template transcription, sulfur relay system, homologous recombination, and mRNA surveillance pathway (Appendix A). Functionally significant terms and pathways of cluster 3 were predominantly associated with ribosome and translation elongation factor activity (Appendix A). Functionally significant terms and pathways of cluster 4 were mainly involved in the GTP biosynthetic ribosome process, ribosome, translation elongation factor activity, circadian rhythm, and RNA transport (Appendix A).

### 3.7. WGCNA Analysis and Identification of Hub Genes

The expressional matrix of HKGs, represented by a TMP index, was transformed and integrated to construct the co-expression network. In this study, a value of β = 12 was selected to ensure a high degree of scale independence (approximately 0.9) and a low mean connectivity (approximately 0) (Figure 6A). Our results of the hierarchical clustering analysis revealed that the correlation expression patterns of HKGs showed the strongest similarities between notochord cartilage and swim bladder, pronephros and epidermis, as well as gallbladder and liver (Figure 6B). These genes were allocated into five co-expression modules; among them, those genes in the grey module could not be classified into any gene module (Figure 6C).

A correlation analysis among these modules indicated that the strongest correlations were among the blue and yellow modules. Conversely, the grey module had low correlations with the other modules (Figure 7A). Remarkably, the grey module had the highest positive correlation with the pronephros tissue, with the highest gene expression levels in the pronephros within the grey module. The turquoise module also exhibited a substantial positive correlation with the brain. In contrast, the muscle group showed a notable negative correlation with both the blue and yellow modules, indicating that the gene expression levels associated with muscle in these modules were relatively low (Figure 7B).

We further extracted those genes and calculated the coefficient of association between gene significance and module membership in the grey module, which was determined to be 0.87 with a *p*-value of 0.00011 (Appendix A). The correlation coefficient between gene significance and module membership within the turquoise module was determined to be 0.87, with a *p*-value of 4.6 × 10^−13^ (Appendix A). The correlation coefficient between gene significance and module membership in the blue and yellow modules were determined to be 0.71 and 0.78, with a corresponding *p*-value of 3.1 × 10^−44^ and 4.8 × 10^−9^, respectively (Appendix A).

Additionally, the module eigengene serves as a representative gene expression profile, encapsulating the expression patterns of all genes within the respective module. The expression profiles of the module eigengenes corresponding to the five identified modules from WGCNA are shown in Figure 8. Compared to other tissues, the eigengene of the grey module showed a notably high expression in pronephros tissue (Figure 8A), whereas the turquoise module was specifically highly expressed in the brain (Figure 8B). The blue module and the yellow module exhibited the highest negative expression of eigengene in muscle, suggesting a significant correlation between muscle and the two modules (Figure 8C,D). Regarding the brown module, the eigengene expression across nine tissues (with the exception of the notochord cartilage and heart) exhibited a notable degree of similarity and uniformity. This finding suggested a robust correlation between the brown module and the other nine tissues, excluding the notochord cartilage and heart (Figure 8E). Additionally, the expression profiles of eigengenes corresponding to the five identified WGCNA modules are consistent with our findings derived from the gene module-gene trait correlation analysis. Subsequent identification of hub genes within the aforementioned modules revealed that only those hub genes that satisfied the established criteria were present in the brown module. The identified hub genes included *rps3a*, *rps7*, *rps23*, *rpl11*, *rpl17*, *rpl27*, *rpl28*, and *g11467* (Figure 8F); except for the g11467 gene, whose function is unknown, the remaining genes were all typical genes encoding ribosomal proteins (RPs), and the GO terms related to those genes included RNA binding and structural constituent of the ribosome [47].

Finally, GO terms and KEGG pathways were enriched for each module. Our results showed that the brown and turquoise modules had significant terms in the GO enrichment. Furthermore, in the KEGG pathway enrichment analysis, alongside the brown and turquoise modules, the blue and yellow modules also had significant pathways in the KEGG enrichment. The biological functions of GO terms in the brown and turquoise modules were mainly related to the ribosome (Appendix A), translation initiation factor activity, and proteolysis involved in the protein catabolic process (Appendix A). Functionally significant pathways in these modules were predominantly associated with ribosome, proteasome, spliceosome, N−glycan biosynthesis, and protein processing in the endoplasmic reticulum (Appendix A).

## 4. Discussion

The Chinese sturgeon, an economically and ecologically important anadromous species with limited distribution in the Yangtze River and the coastal waters of China, has a long lifespan and is recognized as one of the most ancient vertebrates, with a lineage extending over 200 million years. The construction of dams, overfishing, water pollution, and navigation activities have led to a significant decline in the natural populations of Chinese sturgeon. Notably, there has been a complete absence of spawning in the Yangtze River for eight years since 2013. This situation has placed the species at an elevated risk of extinction in the wild [11]. As a result, there has been an increased focus on the artificial propagation and release of stocked individuals to conserve this resource. The advancement of RNA-seq analysis and analytical algorithms has led to incremental progress in the study of gene expression across different tissues. However, the patterns of gene expression derived from extensive datasets encompassing different tissues of Chinese sturgeon remain poorly understood. A thorough investigation of gene expression patterns across different tissues could yield significant insights into the biological functions and regulatory mechanisms for genetic variation in complex traits [48]. Transcriptomes are predominantly used in the field of molecular physiology, and the eleven tissue-specific transcriptomes, along with one comprehensive transcriptome, provided in this study constitute a publicly accessible genomic resource for the study of sturgeons. These transcriptomic resources can be used to elucidate physiological responses to environmental conditions, thereby providing valuable insights for conservation management.

In this study, we constructed a transcriptome atlas of eleven different tissues for Chinese sturgeon. We further determined that some tissues exhibited significant similarities and close associations in expression patterns among some tissues, which may indicate biological identity, coordinated processes, or functional convergence among these tissues. The clustering of tissues suggested shared functional, morphological, physiological, or developmental properties. For example, the high correlation between the liver and the gallbladder is possibly due to the proximity of the right lobe of the liver to the gallbladder, and both of them are the main digestive systems of various teleost [49]. Similarly, the spleen pronephros and epidermis showed highly correlated expression patterns as the epidermis serves as the primary barrier of the body against pathogenic infections [50], and the epidermis, spleen, and pronephros are important organs of the immune system of teleost [51]. The high correlation between the expression patterns of the swim bladder and notochord cartilage can be potentially attributed to their high content of collagen, which has similar structural properties and biological activity to make these tissues primary sources of fish glue derived from sturgeon [52]. These results imply that some gene expression profiles, which may be connected to tissue development, are shared among diverse tissues with comparable physiological roles. This finding is consistent with patterns of multi-tissue gene expression that were previously documented in mammals [53,54].

TSG expression is a recognized biological phenomenon in which the genome produces different transcriptomes across various tissues and cell types. Therefore, the presence of tissue-specific protein-coding transcripts determines the variations in the composition and complexity of transcriptomes among different tissues. Additionally, these transcripts provide insights into the identification of critical pathways and physiological and regulatory processes that are unique to specific tissues [12]. In our current study, we applied tissue specificity indexes, including τ, Counts, and Gini, alongside RNA-seq expression profiles from eleven essential organs of Chinese sturgeon, which facilitated the creation of the first catalog of TSGs and enabled the identification of their associated metabolic processes. In our subsequent analysis, the brain exhibited the most complex transcriptome profile and had the greatest number of tissue-specific protein-coding genes, which constituted 60% of the total identified TSGs.

There are two possibilities that explain the above phenomenon. For one thing, there is neurological complexity. In fact, the brain plays a vital role in integrating environmental signals and coordinating responses. A higher number of TSGs in this organ suggests increased complexity and specialization, which may be necessary for processing sensory information, regulating behavior, and facilitating complex physiological functions. This may indicate that the brain of Chinese sturgeon is particularly adapted to its ecological niche, allowing improved survival and reproduction. Another thing is that there are physiological responses. TSG expression is fundamental to the physiological adaptations of organisms to respond to environmental changes. In the case of Chinese sturgeon, the higher gene expression in the brain may facilitate rapid and effective responses to environmental challenges. This adaptability could be a key factor in the species’ resilience to exogenous factors such as climate change and habitat loss. In summary, the higher number of TSGs in the brain of Chinese sturgeon not only reflects the complexity of its neurological processes but also underscores the significance of gene expression in facilitating physiological adaptations to environmental changes.

This tendency aligns with findings from previous research on various taxa, including domesticated species. Notably, studies on other fish species, such as Atlantic salmon [15], crucian carp [16], and rainbow trout [18], as well as well-established models like birds [55], mice [56], and rats [57], and extending to higher-order mammals including humans [58,59], indicate that the brain consistently expresses the highest levels of tissue-specific transcripts. These findings imply a conserved pattern of tissue-specific expression across major vertebrate taxa and lineages. Functional enrichment analysis revealed that TSGs were predominantly associated with significant biological processes and pathways that are integral to the physiological functions of those tissues. For example, genes specifically expressed in the brain and heart were mainly associated with nerve conduction, muscle contraction, and cardiac rhythm, including voltage-gated potassium channel activity, neurotransmitter, neuroactive ligand−receptor interaction, and cardiac muscle contraction. Those genes that were specifically expressed in the spleen and pronephros are predominantly involved in immune-related biological processes, such as leukocyte transendothelial migration [51].

HKGs are widely and stably expressed across different tissues, serving as internal controls in many biotechnological applications and genomic studies [60,61]. They are necessary for maintaining basal cellular functions. To better understand the HKGs in Chinese sturgeon, we screened 787 HKGs and observed that their expression levels varied across different tissues. Functionally significant terms and pathways based on HKGs were primarily associated with elongation factor activity, ribosome, translation initiation factor activity, proteasome core transcription complex, proteolysis involved in protein synthesis and catabolic process, and RNA transport. All of them are related to maintaining basic cell functions and energy metabolism, which is consistent with previous reports [56,62]. Specifically, a cell’s structural foundation is provided by proteins, which also carry out the enzymatic processes to support DNA replication and energy production. Proteins make up the hormones and growth factors that permit organs to communicate with one another, as well as the receptors and signaling bridges that connect extracellular inputs to intracellular activity [63]. Our results indicated that protein translation factors and other so-called housekeepers have important roles in controlling the cell growth and apoptosis of Chinese sturgeon.

However, a significant limitation in the functional interpretation of HKGs arises from the fact that each gene is analyzed independently. In reality, no gene is capable of executing biological functions independently. Most genes are part of functional modules and complex networks of interacting proteins. WGCNA is frequently used to develop co-expression modules and to identify hub genes within these modules by determining the correlation patterns among genes based on the analysis of the similarities in gene expression profiles [41]. Our findings from the TSG analysis indicated that, with the exception of the brain, the number of genes exhibiting specific expression in other tissues is relatively limited. Thus, we constructed five co-expression modules with HKGs from eleven tissues of Chinese sturgeon by WGCNA and identified hub genes in each module. In particular, a total of eight hub genes were identified within the brown module, but no additional hub genes were detected in the other modules. Among the identified eight hub genes, the *g11467* gene was classified as one with an unknown function, while the remaining seven genes (*rps3a*, *rps7*, *rps23*, *rpl11*, *rpl17*, *rpl27*, and *rpl28*) were all RP (ribosomal protein) coding genes.

RPs are essential for ribosome biogenesis and protein synthesis, producing all proteins necessary for cellular growth and maintenance, and play a significant role in various developmental processes. In eukaryotic organisms, the synthesis of ribosomes is a multifaceted process. Briefly, each 80S ribosome is comprised of two ribonucleoprotein subunits. The larger 60S subunit consists of the 28S, 5.8S, and 5S ribosomal RNAs (rRNAs) along with 47 associated ribosomal proteins L (RPLs). Instead, the smaller 40S subunit contains the 18S rRNA and 33 ribosomal proteins S (RPSs) [64]. The cellular concentrations of some RP transcripts show variations in response to growth, development, and certain tumors [65]. Furthermore, numerous RPs are thought to play significant roles in a variety of additional cellular processes, commonly referred to as extraribosomal functions [66]. The expression profiles of *rp* genes across various tissues and developmental stages have been reported in multiple species. In tilapia, the *rp* genes were found to be expressed across all examined tissues, but the levels of expression varied among different tissues [67]. In *Xenopus tropicalis*, during stages 18 to 22, 13 *rps* genes were found to be expressed in the neural tube, developing brain, and migrating neural crest, and *rps7* exhibited a significantly higher level in the somites compared to other *rps* mRNAs [68]. In channel catfish (*Ictalurus punctatus*), Senegalese sole (*Solea senegalensis*), and Atlantic halibut (*Hippoglossus hippoglossus*), *rp* genes have been demonstrated to exhibit ubiquitous expression at comparable levels across various tissues [69,70].

In our present study, seven *rp* genes were found to be expressed in all eleven tissues of Chinese sturgeon, but their expression levels differed among diverse tissues, indicating their critical functions in multiple physiological processes. Moreover, it has long been suggested that any hereditary defect in ribosome production would result in embryonic lethality. Furthermore, various human congenital disorders have been linked to mutations in certain genes encoding ribosome biogenesis factors. For example, Diamond–Blackfan anemia (DBA) is a common feature of ribosomopathies, which is associated with lesions in a minimum of 15 different RPs (RSP7, RPS10, RPS17, RPS19, RPS24, RPS26, RPS27, RPS29, RPL5, RPL11, RPL15, RPL26, RPL27, RPL31, and RPL35A) [68,71,72]. These abnormalities impact the processing of pre-rRNA and the assembly of the small ribosomal subunit or large ribosomal subunit, often leading to bone marrow failure. Additionally, some patients frequently exhibit craniofacial anomalies, as well as defects in growth and limb development [71]. The expression of all 15 genes involved in ribosome production showed a significant correlation with the cranial neural crest and developing brain [68]. Other major ribosomopathies related to the hub genes obtained in this study mainly include familial encephalopathy with neuroserpin inclusion bodies (FENIB), associated with RPS3A; brachycephaly, trichomegaly, and developmental delay and autism spectrum disorder, associated with RPS23; and epilepsy, idiopathic generalized 14 and coenzyme Q10 deficiency, associated with RPL28 [47]. The related pathways among the hub genes identified in this study were viral mRNA translation and nervous system development [47]. Knockdown of a substantial cohort of *rps* and *rpl* genes in developing zebrafish has indicated that different axis defects and central nervous system abnormalities can be specifically attributed to individual RPs, which implies that ribosome-related genes may play a critical role in tissue-specific development [73]. Our results of this study also support that RP coding genes played an important role in tissue differentiation and nerve development in Chinese sturgeon. In addition, this research lays a foundation for understanding the fate of various genes following the whole-genome duplication in Chinese sturgeon, thereby enhancing our knowledge of the evolution and expression of duplicated genes.

Taken together, expression profiles of HKGs andTSGs based on transcriptomics were obtained using combined bioinformatics approaches, including functional annotation, gene enrichment analysis, pathway analysis, and network construction and visualization. Our valuable findings in this study may serve as a significant reference for further research. Understanding the regulation mechanism of Chinese sturgeon allows for the targeted management of conservation strategies. In the case of Chinese sturgeon, scientific prerequisites must be established in order to complete an action plan for its resource protection, which includes strengthening environmental assessments, paying greater attention to forecasting prevention success, and monitoring progress during measure implementation. These measures primarily involve two pathways: (1) Improving habitat management: insights gained from TSG expression can deepen our understanding of the ecological requirements for Chinese sturgeon. This information is vital for effective habitat restoration and management efforts, ensuring that conservation strategies align with the species’ biological needs. (2) Informing policy and advocacy: the findings from our current study can provide scientific evidence to bolster conservation policies and initiatives. By emphasizing the ecological and evolutionary significance of Chinese sturgeon, we advocate for stronger protections and additional resources for its conservation, which will benefit the prosperity and development of this extremely important species.

## 5. Conclusions

In this study, the transcriptomic profiles of eleven vital tissues of Chinese sturgeon were explored. We found that the brain had the highest number of expressed protein-coding genes. A total of 787 housekeeping genes were identified, and the number of tissue-specific genes varied from 25 in the liver to 2073 in the brain. The signaling pathways associated with tissue-specific genes suggested that the exclusively expressed genes were likely involved in the regulation of distinct physiological processes associated with specific tissues, such as nerve conduction, muscle function, and cardiac rhythm regulation. In addition, eight hub genes were predicted in the brown module. Most of the hub genes were *rps* and *rpl* genes, which may play important roles in tissue differentiation and nerve development in Chinese sturgeon. The valuable transcriptome datasets will increase the existing information on Chinese sturgeon in public data repositories and will provide valuable insights for investigations into the biological development and processes of various tissues in Chinese sturgeon.

## Figures and Tables

**Figure 1 animals-14-03357-f001:**
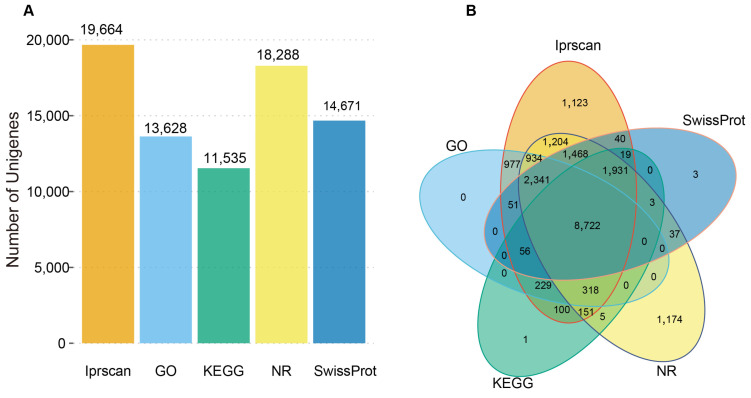
Gene annotation of the Chinese sturgeon transcriptomes. (**A**) Bar graph of the number of genes annotated to the Iprscan, GO, KEGG, NR, and SwissProt databases. (**B**) Venn diagram of transcripts against the Iprscan, GO, KEGG, NR, and SwissProt databases.

**Figure 2 animals-14-03357-f002:**
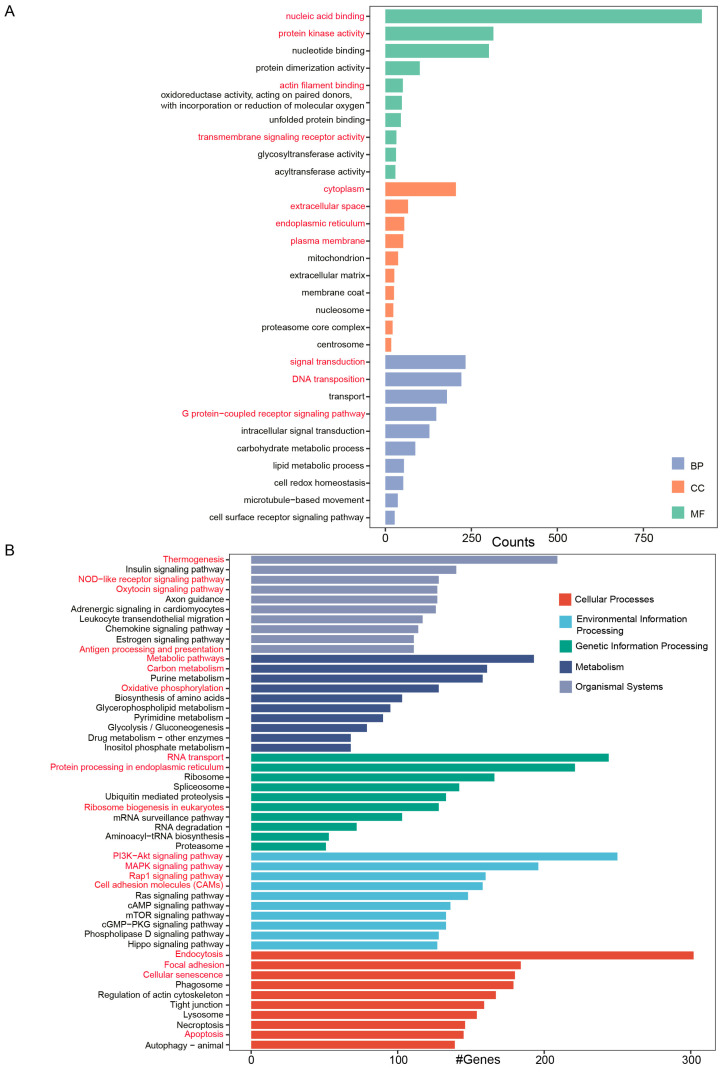
Functional annotation of the Chinese sturgeon transcriptomes. (**A**) GO terms for the enriched transcripts. (**B**) KEGG pathway categories of the enriched transcripts.

**Figure 3 animals-14-03357-f003:**
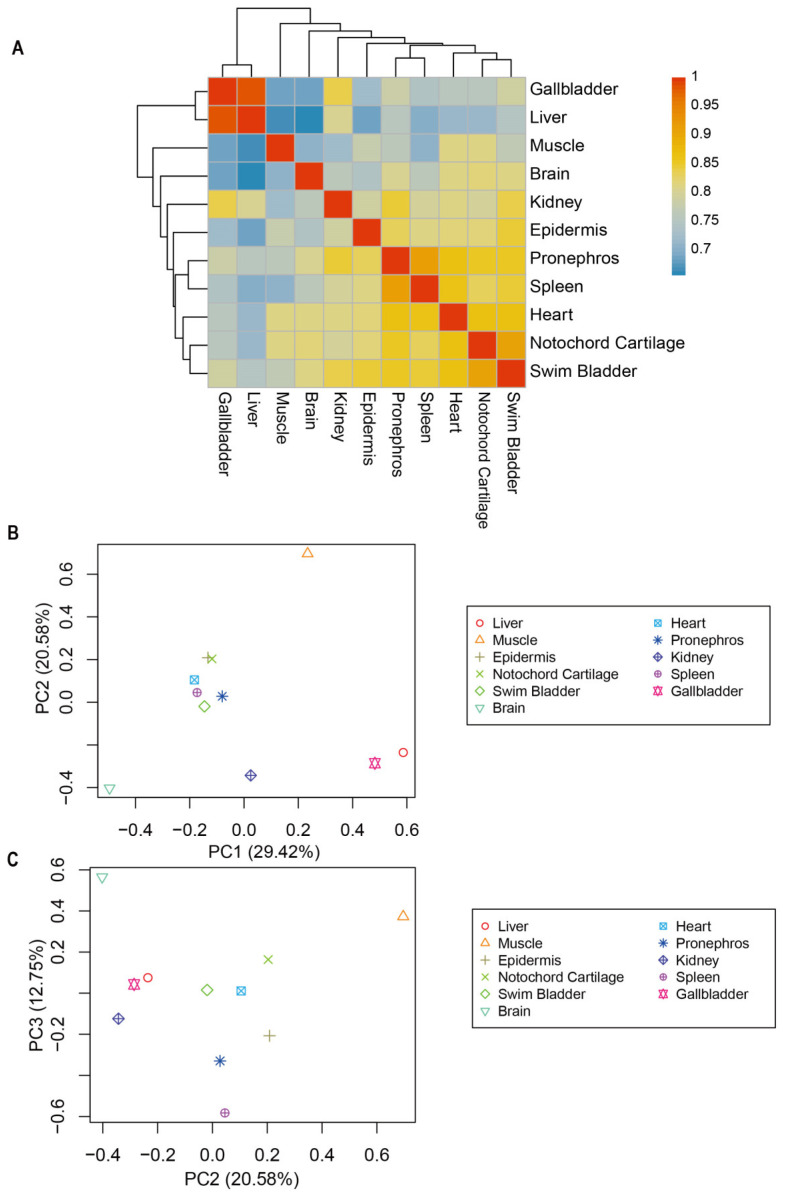
Correlation of eleven tissues of Chinese sturgeon. (**A**) Pearson’s correlation analysis for comparisons among the eleven tissues. (**B**) PCA plot of all tissues with the first two dimensions, PC1 and PC2. (**C**) PCA plot of all tissues with the second two dimensions, PC2 and PC3.

**Figure 4 animals-14-03357-f004:**
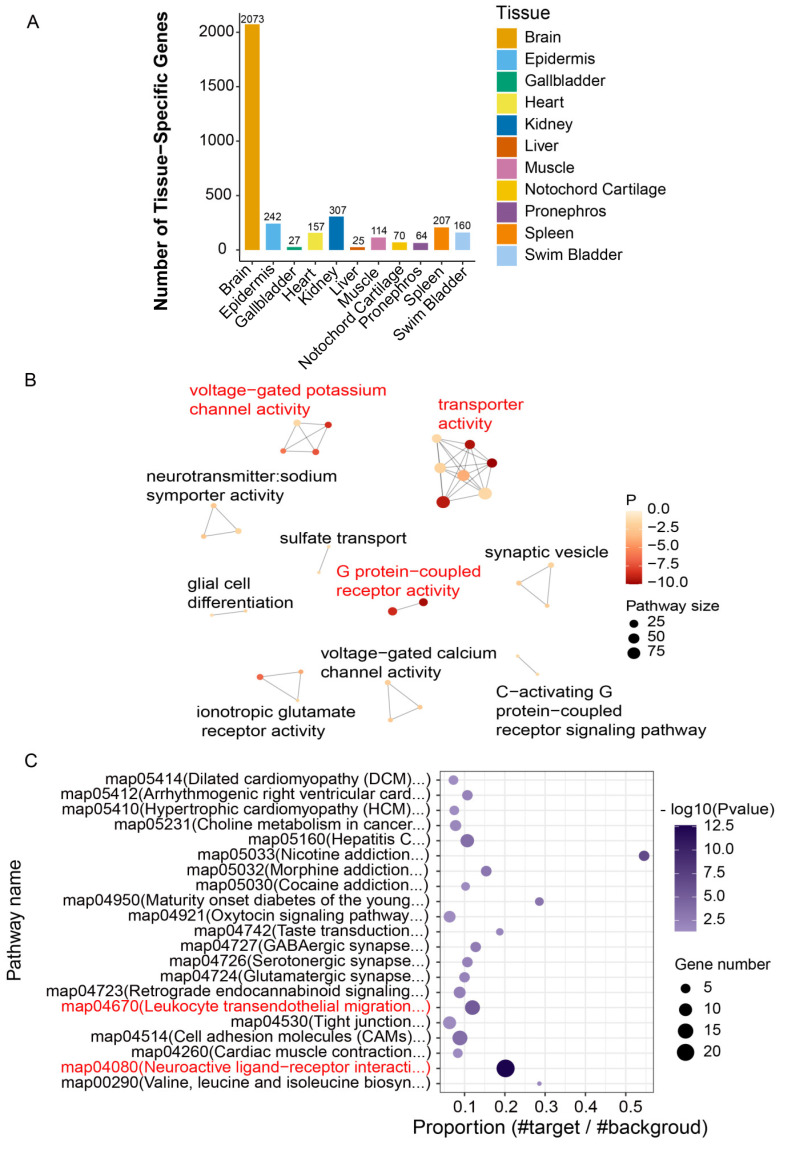
Statistics of TSG number and functional enrichment. (**A**) Histogram of the TSG number. (**B**) Network diagram depicting the significantly enriched GO terms for TSGs. (**C**) Bubble chart of significantly enriched KEGG pathways for TSGs.

**Figure 5 animals-14-03357-f005:**
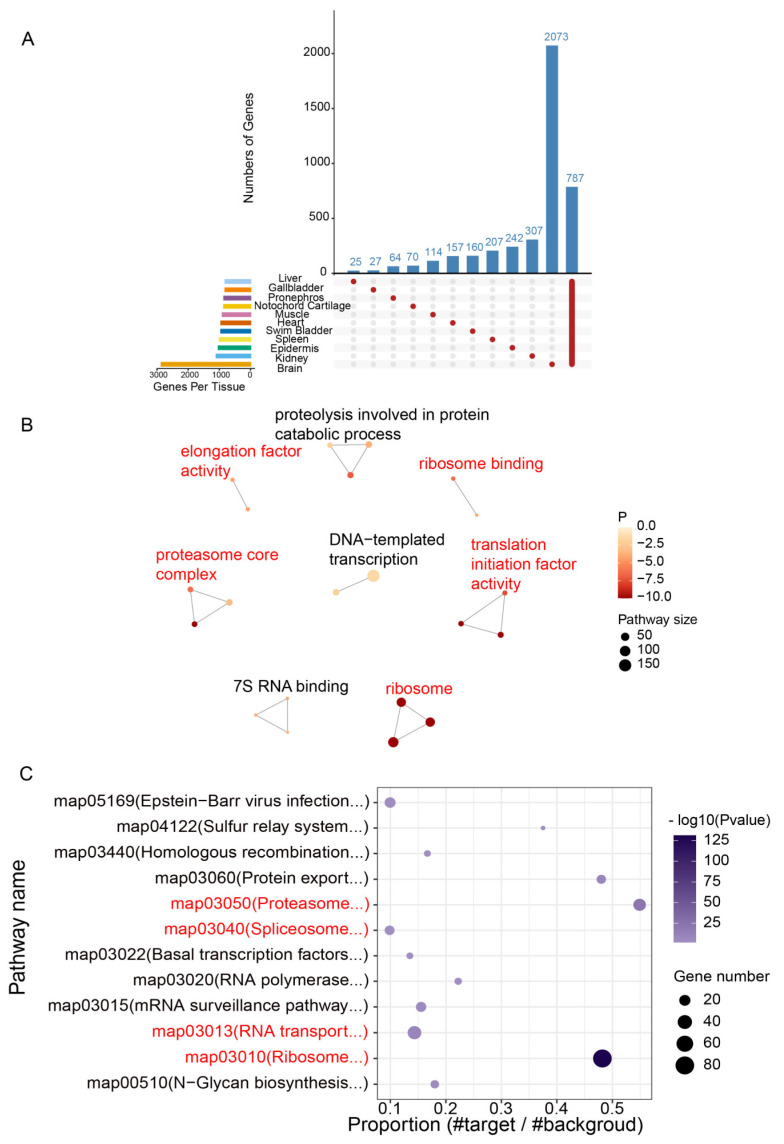
Identification of HKGs and functional enrichment analysis. (**A**) UpSet plot of shared and unique genes from each of the eleven tissue-specific transcriptomes of Chinese sturgeon. (**B**) Network diagram showing the significantly enriched GO terms for HKGs. (**C**) Bubble chart of significantly enriched KEGG pathways for HKGs.

**Figure 6 animals-14-03357-f006:**
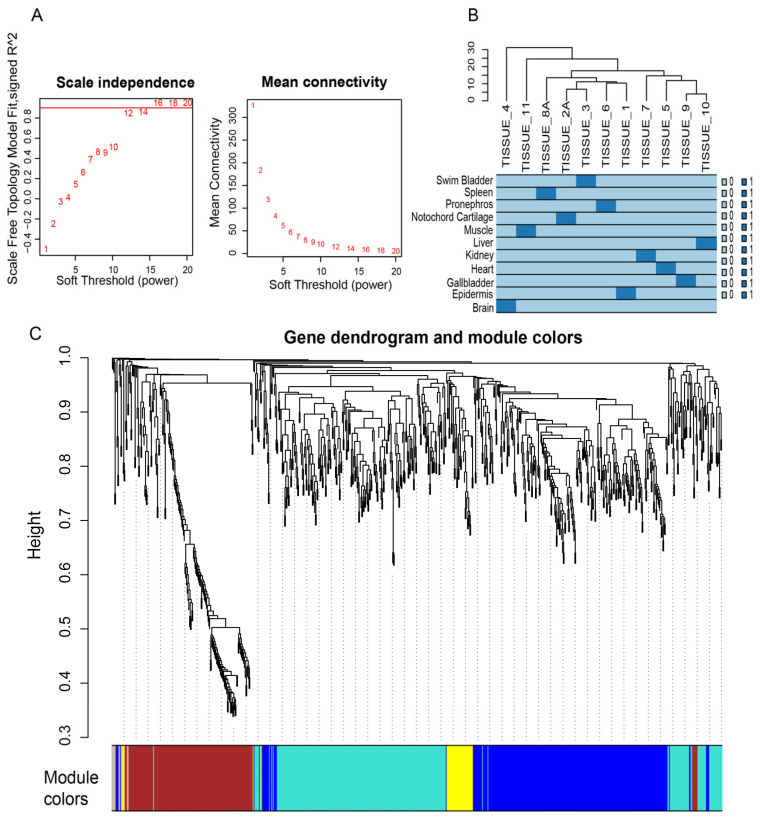
Construction of a weighted gene co-expression network based on the HKG TPM expression matrix. (**A**) An examination of the scale-free topology model fit index for soft threshold power (β) and the mean connectivity for soft threshold powers. The horizontal axis represents β, and the vertical axis on the left represents the corresponding log(k) and log(p(k)) in the network. The higher the square of the correlation coefficient, the closer the network is to a scale-free distribution. The adjacency matrix was defined using soft thresholds with β = 12. (**B**) Hierarchical clustering diagram of the eleven tissues of Chinese sturgeon. (**C**) A dendrogram showing the hierarchical clustering based on the TOM matrix derived from the gene expression data. Various colors correspond to distinct modules, in which the grey module indicates no co-expression among the genes.

**Figure 7 animals-14-03357-f007:**
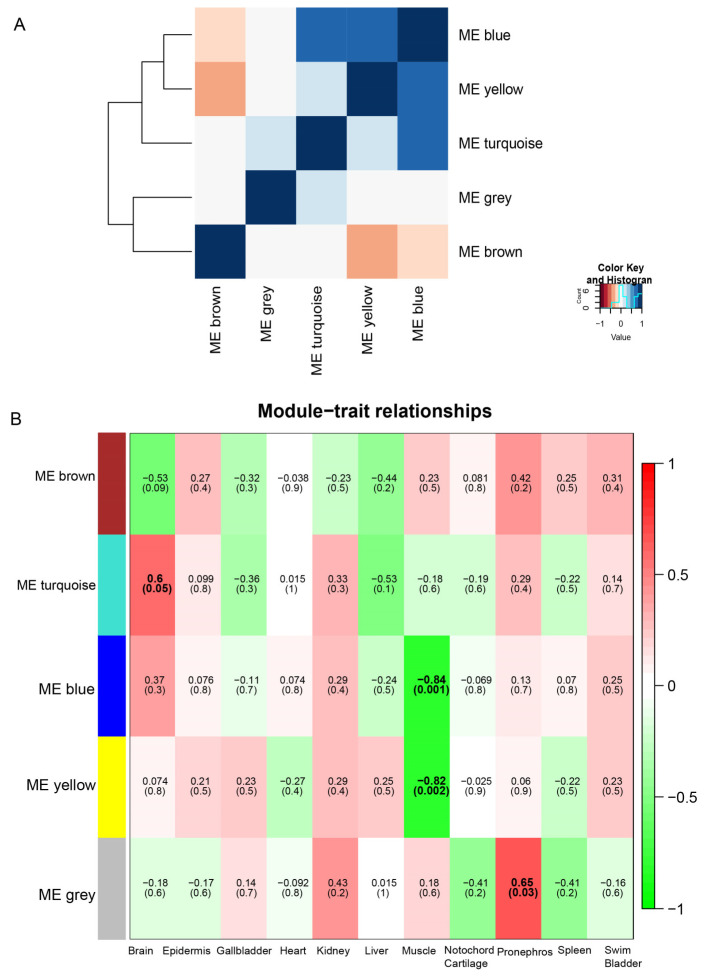
Module and eigengene network plot. (**A**) Analysis of the correlation between modules. (**B**) A heatmap visualization showing the adjacencies within the eigengene network. The diagram shows modules in the rows and phenotypes (tissues) in the columns, with colored blocks indicating the association between them. The heatmap used a color palette in which green represents low adjacency, indicating a negative correlation, while red represents high adjacency, indicating a positive correlation. The numerical values within the colored blocks denote the Pearson correlation coefficients, while those in parentheses are *p*-values. Blocks with *p* ≤ 0.05 were deemed to show statistically significant correlations.

**Figure 8 animals-14-03357-f008:**
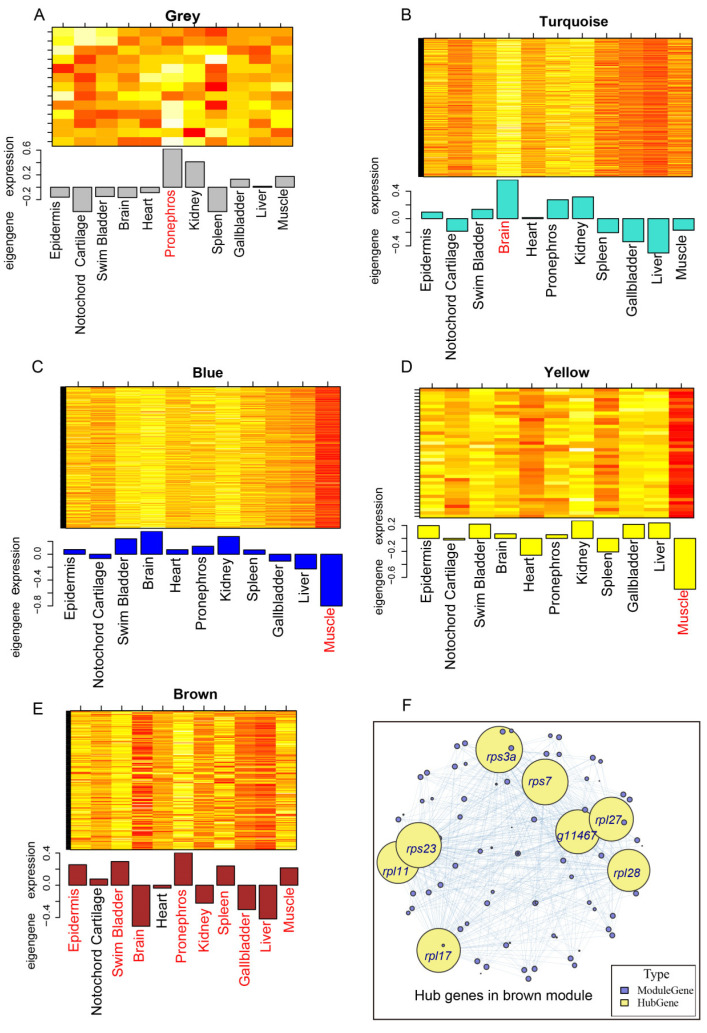
Expression profiles of eigengenes for the identified WGCNA modules and hub genes in the brown module. (**A**) Expression profiles of eigengenes for the blue module (MC = −0.84, 280 genes). (**B**) Expression profiles of eigengenes for the grey module (MC = 0.65, 13 genes). (**C**) Expression profiles of eigengenes for the turquoise module (MC = 0.6, 287 genes). (**D**) Expression profiles of eigengenes for the yellow module (MC = −0.82, 39 genes). (**E**) Expression profiles of eigengenes for the brown module (MC = −0.53, 171 genes). The colors directly correspond to the names assigned to the modules, and each bar within the module represents a different tissue. (**F**) Hub genes are identified in the brown module. The genes within the module are organized based on their within-module level in a descending manner, with the top 5% of genes identified as the hub genes (highlighted in yellow in the diagram). The node size reflects the level of connectivity between the node and neighboring genes within the module.

**Table 1 animals-14-03357-t001:** Details of transcriptome sequencing of *Acipenser sinensis* using a PacBio Sequel platform.

Type	Item	Number
Polymerase reads	Polymerase read bases (G)	83.45
	Read number	806,926
	Average length (bp)	103,422
	N50 length (bp)	193,380
Subreads	Subread bases (G)	79.88
	Subread number	46,605,450
	Average length (bp)	1714
	N50 length (bp)	2118
CCS reads	Read number	318,019
	Average length (bp)	2200
	N50 length (bp)	2640
FLNC reads	Read number	108,170
	Average length (bp)	2166
	N50 length (bp)	2746
	FLNC/CCS	34.01%
Unigenes	Read number	25,434
	Min length (bp)	307
	Max length (bp)	9515
	Average length (bp)	2691
	N50 length (bp)	3195

## Data Availability

The transcriptome sequencing data of *Acipenser sinensis* presented in this study have been deposited in the Genome Sequence Archive (GSA) database of National Genomics Data Center (NGDC) under the accession number CRA018927, with the associated Bioproject number PRJCA030015. Further details regarding data availability can be found at https://ngdc.cncb.ac.cn/gsa/s/TQoE0eKh (accessed on 2 November 2024).

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
