# Peer review of "Expression Profiles of Housekeeping Genes and Tissue-Specific Genes in Different Tissues of Chinese Sturgeon (Acipenser sinensis)"

_animals, 2024, doi:10.3390/ani14233357_

Round 1
Reviewer 1 Report
Comments and Suggestions for Authors
In this article, the authors give gene expression profiles across different tissues in Acipenser sinensis.
Major issues
Thank you for your interest in conducting WGCNA analysis. However, I disagree the current number of samples (11) is insufficient to perform a reliable WGCNA analysis. Typically, WGCNA requires a larger sample size to produce robust and meaningful results, as it is highly sensitive to the number of input samples for accurate network construction and module detection. To improve the reliability of the analysis, I recommend increasing the sample size to at least 15-20, depending on the complexity of the data. This will help ensure that the results are more robust and biologically meaningful.
I noticed that the number of biological replicates is not clearly stated. Replicates are crucial in RNA-seq analysis to ensure the statistical power and reliability of the results, especially when assessing gene expression differences.Transcript annotation of Chinese sturgeon (Acipenser sinensis) using Iso-seq and RNA-seq data
The discussion section does not fully support all of your results. You should focus more on balancing the presentation of results with a corresponding discussion to ensure coherence and depth.
The significant divergence were found between your results and those already published in the study "Transcript annotation of Chinese sturgeon (Acipenser sinensis) using Iso-seq and RNA-seq data." Given that this earlier research provides a detailed and widely accepted reference, the substantial discrepancies in your findings require more rigorous justification and analysis to establish their validity.
Comments on the Quality of English LanguageMinor editing
Author Response
Comments 1: Thank you for your interest in conducting WGCNA analysis. However, I disagree the current number of samples (11) is insufficient to perform a reliable WGCNA analysis. Typically, WGCNA requires a larger sample size to produce robust and meaningful results, as it is highly sensitive to the number of input samples for accurate network construction and module detection. To improve the reliability of the analysis, I recommend increasing the sample size to at least 15-20, depending on the complexity of the data. This will help ensure that the results are more robust and biologically meaningful.
Response 1: Thanks for your insightful advice regarding the methodology. I fully appreciate your point that a sample size of 15-20 is generally more suitable for a WGCNA analysis, as it can indeed enhance the reliability of results by reducing errors in correlation matrix construction.
However, I would like to mention that our current study differs from typical transcriptomic analysis, which often includes multiple conditions with replicates. In those cases, replicates help to minimize variance among samples by reducing the covariance of gene expression. As a result, a larger sample size is usually required to capture biological variability.
In our current research, we collected samples from various tissues, which naturally exhibit substantial differences in gene expression patterns. Although we utilized a smaller number of samples for our WGCNA, we believe that this approach still allowed us to glean valuable insights due to the significant biological variability among these different tissues.
Thank you once again for your constructive feedback, and we will certainly take your recommendations into consideration for our future studies.
Comments 2: I noticed that the number of biological replicates is not clearly stated. Replicates are crucial in RNA-seq analysis to ensure the statistical power and reliability of the results, especially when assessing gene expression differences. Transcript annotation of Chinese sturgeon (Acipenser sinensis) using Iso-seq and RNA-seq data
Response 2: Thank you for your valuable opinion regarding the number of biological replicates. I truly appreciate your emphasis on the importance of replicates in RNA-seq analysis for ensuring the statistical power and reliability of results. In our present study, we focused on the Chinese sturgeon (Acipenser sinensis), a species that is classified as highly protected. The samples used in our current research were obtained from a single approximately one-year-old individual. We collected RNA from eleven representative tissues throughout its body. To comprehensively capture the gene expression sequences from these tissues, we performed Iso-seq on mixed samples. Additionally, to further investigate gene expression variations across these different tissues, we conducted individual RNA-seq analyses for each sample.
Based on your feedback, we have revised the description of sample collection for clarity. Please refer to Lines 171-186 for the updated content.
Comments 3: The discussion section does not fully support all of your results. You should focus more on balancing the presentation of results with a corresponding discussion to ensure coherence and depth.
Response 3: Thank you for your insightful feedback. Your advice is valuable for improving the quality of our manuscript. In this revision, we have reorganized and enhanced the discussion section to align more closely with the order of the presented results. Additionally, we have incorporated your comments along with those of other reviewers to strengthen our literature citations. See more details in lines 671-695; 708-722; 738-753, and 811-835 of our revised manuscript.
Comments 4: The significant divergence was found between your results and those already published in the study "Transcript annotation of Chinese sturgeon (Acipenser sinensis) using Iso-seq and RNA-seq data." Given that this earlier research provides a detailed and widely accepted reference, the substantial discrepancies in your findings require more rigorous justification and analysis to establish their validity.
Response 4: Thanks for your instructive comments. While there are indeed discrepancies between our findings and those of the previous study "Transcript annotation of Chinese sturgeon (Acipenser sinensis) using Iso-seq and RNA-seq data," it is important to note that our samples do not overlap with those in the earlier research except for the swim bladder and liver. Additionally, the age of the specimens collected in our present study differs from those in the previous report. Given the spatiotemporal specificity of gene expression, we believe that our current research provides a more comprehensive and detailed description of gene expression patterns in this species. By the way, we provide a brief explanation about the substantial discrepancies in lines 106-107, and 183-186.

Reviewer 2 Report
Comments and Suggestions for Authors
Dear Authors,
Thank you for your extensive and detailed work. I thoroughly enjoyed reading your manuscript, and I must say it has been a while since I have reviewed a paper that is so well-prepared in terms of technical, formal, and scientific standards. Congratulations on producing such a high-quality manuscript. Below, I have noted a couple of points that may further enhance your study:
- It seems that only one fish specimen was collected from the wild. However, there is insufficient information in section 2.1 regarding how the fish's health was assessed, what criteria were used to determine if it was healthy, and how a transcriptomic analysis from a single fish represents the entire population.
- Since you have genomic data, it would be valuable to conduct an ecological distribution analysis. This could provide broader insights into the distribution of the fish species in different ecological environments, thus adding a valuable dimension to the provisional international distribution of the species investigated.
Sincerely.
Author Response
Thank you for your extensive and detailed work. I thoroughly enjoyed reading your manuscript, and I must say it has been a while since I have reviewed a paper that is so well-prepared in terms of technical, formal, and scientific standards. Congratulations on producing such a high-quality manuscript. Below, I have noted a couple of points that may further enhance your study:
Comments 1: It seems that only one fish specimen was collected from the wild. However, there is insufficient information in section 2.1 regarding how the fish's health was assessed, what criteria were used to determine if it was healthy, and how a transcriptomic analysis from a single fish represents the entire population.
Response 1: Thanks for your generous feedback and the good question. Concerning health assessment of the fish that was collected from the wild, we recognize the importance of providing additional clarification in section 2.1. In fact, before collection we conducted a thorough health assessment based on some established criteria, which included a visual inspection for signs of disease and a behavioral assessment, such as observing swimming patterns. These evaluations were essential in ensuring that the fish represented a healthy individual within the examined population.
We understand the limitations associated with conducting transcriptomic analysis on a single specimen. Our goal was to lay the groundwork for understanding gene expression patterns in Chinese sturgeon, and we hope that the insights gained in this study will inform future investigations that incorporate larger sample size and more comprehensive population analyses.
We revised section 2.1 to include these details for enhanced clarity and transparency. Please see more details in the lines 171-186 of the revised manuscript.
Comments 2: Since you have genomic data, it would be valuable to conduct an ecological distribution analysis. This could provide broader insights into the distribution of the fish species in different ecological environments, thus adding a valuable dimension to the provisional international distribution of the species investigated.
Response 2: Thank you for your valuable suggestion regarding the ecological distribution analysis. Currently, we are still in the process of collecting species distribution data for Chinese sturgeon. Given their migratory nature, continuous observation of their movement patterns is necessary, while ensuring the health and well-being of the individuals being studied. This presents certain challenges, but we intend to address these issues in our future research.

Reviewer 3 Report
Comments and Suggestions for Authors
Why random hexamer primer was chosen for first-strand cDNA synthesis, as this can impact the representation of different transcripts. Are biological replicates were included in the analysis. Why the est_method salmon parameter was chosen over other methods? What is the significance of using PacBio Sequel technology for transcriptome analysis? Advantages
What is the Significance of the Chinese sturgeon and its conservation status?
Clearly describe the methodologies used and specify the techniques used?
Break down long sentences into shorter ones to improve readability and comprehension?
How these findings can inform conservation strategies for the Chinese sturgeon.
Start with a statement about the ecological and evolutionary significance of the Chinese sturgeon?
Summarize key statistics about population decline?
Add latest reference?
How specific infrastructure projects i.e. Gezhouba Dam have impacted migration patterns and spawning grounds?
Review for grammatical consistency and clarity, particularly in longer sentences.?
What is the age or size of the juvenile sturgeon used for sampling
Ethical approval?
whyTRIzol was chosen?
Define technical terms "SMRT" (Single Molecule Real-Time) sequencing
Discuss any measures taken to avoid RNA degradation or contamination during sample collection and processing.Are any measures taken to prevent contamination during library preparation & sequencing, as RNA is highly susceptible to degradation.
Summarize key findings at the end. Add significance level of p < 0.05 in table 1?
Population Decline Details: Include specific statistics or studies?
Discuss any successes or challenges faced in artificial propagation efforts?How understanding gene expression patterns can directly contribute to conservation strategies and management practices?
Discuss the implications of the variation in tissue-specific genes, particularly why brain tissue has a higher number compared to others. Expand on the significance of tissue-specific gene expression by discussing its implications for understanding physiological adaptations to environmental changes?
How the comparisons enhance our understanding of evolutionary biology in sturgeons?
Start with a concise summary of the most significant findings before the details.?
Expand on the significance of the identified functional pathways and biological processes associated with housekeeping genes, perhaps linking them to broader physiological contexts.?
Emphasize the importance of making your transcriptome dataset publicly available?
End with a strong concluding statement that summarizes the overall significance of your work and its potential impact on future studies related to aquaculture and conservation biology.?

Author Response
Comments 1: Why random hexamer primer was chosen for first-strand cDNA synthesis, as this can impact the representation of different transcripts. Are biological replicates were included in the analysis. Why the est_method salmon parameter was chosen over other methods? What is the significance of using PacBio Sequel technology for transcriptome analysis? Advantages
Response 1: Thanks for your good questions, which are answered as follows.
Choice of Random Hexamer Primers: We opted for random hexamer primers during first-strand cDNA synthesis to achieve comprehensive coverage of the transcriptome. This approach enables us to amplify a wide variety of transcripts, including those expressed at lower levels. We have revised the description of the library construction for clarity. Please refer to lines 227-231 for the updated content. While oligo(dT) primers would primarily focus on polyadenylated mRNA, random hexamers help to ensure a broader representation of both coding and non-coding RNAs.
Inclusion of Biological Replicates: I must clarify that our study utilized a single Chinese sturgeon specimen, approximately one-year-old, for sampling of various tissues.
Choice of Est_method Salmon Parameter: We chose the "est_method" parameter as "salmon" due to its efficiency in quantifying transcript abundances from RNA-seq data. Salmon utilizes advanced statistical models that allow for accurate estimation of expression levels, which we found advantageous for our dataset. Its ability to perform quasi-mapping enables us to quantify transcripts without needing extensive alignment. We have revised the description of the quantifying transcript abundance in the section 2.5 for clarity. Please refer to Lines 289-293 for the revised version.
Significance of PacBio Sequel Technology: We believe that utilizing PacBio Sequel technology in our transcriptomic analysis provides several significant advantages. The long-read sequencing capabilities of this technology enhance us to assemble complex transcripts and identify isotypes. Additionally, the high accuracy of PacBio reads improves our resolution of transcript structures, including alternative splicing events and novel transcripts, thus offering a more comprehensive understanding of the transcriptomes.
Comments 2: What is the Significance of the Chinese sturgeon and its conservation status?
Response 2: Thanks for your question. In fact, Chinese sturgeon (Acipenser sinensis), a species with important scientific and economic values, has been listed as a First Class Protected Animal by the China Government in 1989. It was most recently assessed for the IUCN Red List of Threatened Species in 2019, where it is classified as Critically Endangered under criteria A2bc. Furthermore, in a recent assessment in 2022, it continues to be recognized as a threatened species (Wei Q., 2022. Acipenser sinensis [amended version of 2022 assessment]. The IUCN Red List of Threatened Species 2022: e.T236A219152605. https://dx.doi.org/10.2305/IUCN.UK.2022-2.RLTS.T236A219152605.en). We added these contents in the revised manuscript (see more details in lines 65-71).
Comments 3: Clearly describe the methodologies used and specify the techniques used?
Response 3: Yes, we have thoroughly modified and improved the Methods section. See more details in lines 171-186; 218-223; 225-232; 289-298; 308-324.
Comments 4: Break down long sentences into shorter ones to improve readability and comprehension?
Response 4: Thanks for your instructive advice. We have split many long sentences, such as those in lines 315-324, 339-347, 389-392, and 510-520.
Comments 5: How these findings can inform conservation strategies for the Chinese sturgeon.
Response 5: Thanks for your thoughtful question. I would like to address this from two perspectives:
Improving Habitat Management: Insights gained from tissue-specific gene expression can enhance our understanding of the ecological requirements of the Chinese sturgeon. This information is crucial for habitat restoration and management efforts, helping ensure that conservation strategies are well-aligned with the biological needs of the target species.
Informing Policy and Advocacy: The findings from our present study can serve as scientific evidence to support conservation policies and initiatives. By highlighting the ecological and evolutionary significance of the Chinese sturgeon, we can better advocate for stronger protections and resources dedicated to its conservation.
See more details in lines 826-835.
Comments 6: Start with a statement about the ecological and evolutionary significance of the Chinese sturgeon?
Response 6: Yes, it is done (see more details in the lines 22-23).
Comments 7: Summarize key statistics about population decline?
Response 7: Thank you for your nice question. I would like to summarize the key statistics regarding the decline of the Chinese sturgeon population from two perspectives.
Population Statistics: The current estimates indicate that the wild population of the Chinese sturgeon is extremely low.
IUCN Status: The Chinese sturgeon is classified as "Critically Endangered" by the International Union for Conservation of Nature (IUCN), highlighting the significant threats to its survival.
Please find more details in lines 61-71.
Comments 8: Add latest reference?
Response 8: Thanks for your instructive advice. Yes, we updated some recent references such as ref 1, 4, 53, 54, 56, 60, 61, 63 and 64 in the revised manuscript. The changes were highlighted in blue.
Comments 9: How specific infrastructure projects i.e. Gezhouba Dam have impacted migration patterns and spawning grounds?
Response 9: Thanks for your instructive question. As we know, the series of dams constructed along the Yangtze River is a significant factor contributing to the decrease in the wild population of Chinese sturgeon, adversely impacting its natural reproductive processes and serving as a primary agent of harm to the species. Analyzing the habitat requirements throughout the life cycle of Chinese sturgeon reveals that the obstruction created by the Gezhouba Dam has resulted in a diminished number of viable breeding populations and a reduction in the availability of spawning grounds.
The specific reasons can be attributed to the damming by Gezhouba Dam, which has resulted in a reduction of migration distance, a decrease in survival rates, and a postponement of the breeding season. This situation is identified as the principal cause of the decline in Chinese sturgeon resources. We have added this content in the lines 76-81 of the revised manuscript.
Comments 10: Review for grammatical consistency and clarity, particularly in longer sentences?
Response 10: Thanks for your instructive advice. Yes, we have thoroughly revised this issue in our present version of manuscript.
Comments 11: What is the age or size of the juvenile sturgeon used for sampling?
Response 11: The specimen of Chinese sturgeon utilized in this research was one-year-old juvenile. We added this detail in the revised manuscript (line 171).
Comments 12: Ethical approval?
Response 12: Yes, we added the animal ethics approval in the revised manuscript (in lines 874-876).
Comments 13: Why TRIzol was chosen?
Response 13: In fact, we selected TRIzol for RNA extraction because it is a highly effective reagent for isolating high-quality RNA, particularly from tissues that may contain complex mixtures of biomolecules. TRIzol allows for the simultaneous extraction of RNA, DNA, and proteins, which can be used for comprehensive downstream analyses. Additionally, its well-established protocols have been widely validated in the literature, ensuring reliable results in transcriptomic studies.
Comments 14: Define technical terms "SMRT" (Single Molecule Real-Time) sequencing.
Response 14: "SMRT" stands for Single Molecule Real-Time sequencing, which is a next-generation sequencing technology developed by Pacific Biosciences Inc. This method allows for the real-time observation of DNA synthesis by capturing the activity of DNA polymerases on single DNA molecules. SMRT sequencing produces long-read sequences, enabling more accurate assembly of genomes and improved resolution of complex genomic regions. This technology is particularly beneficial for studying structural variants and for de novo genome assembly. We have added this content in the revised manuscript (see more details in lines 218-223).
Comments 15: Discuss any measures taken to avoid RNA degradation or contamination during sample collection and processing. Are any measures taken to prevent contamination during library preparation & sequencing, as RNA is highly susceptible to degradation.
Response 15: Yes. To prevent RNA degradation and contamination during sample collection and processing, several measures were performed. First, all equipment and instruments were thoroughly cleaned and sterilized prior to use. During sample collection, tissues were transferred into the centrifuge tube with RNAlater and promptly stored in liquid nitrogen to preserve RNA integrity. Additionally, samples were kept on ice throughout the collection process to minimize any temperature-related degradation.
In the laboratory, RNA extraction was performed in a dedicated area, ensuring that environmental contaminants were minimized. Gloves and face masks were worn at all times, and materials were handled using RNase-free techniques.
For library preparation and sequencing, rigorous protocols were followed to prevent contamination. All reagents and instruments used were certified RNase-free. Work surfaces were treated with RNA-decontamination solutions, and laminar flow hoods were utilized during library preparation to provide a sterile environment. Furthermore, all samples were processed as quickly as possible to limit exposure to potential contaminants. By following these protocols, we aimed to safeguard the RNA integrity throughout the entire process.
We added some details in lines 181-183, and 214-216.
Comments 16: Summarize key findings at the end. Add significance level of p < 0.05 in table 1?
Response 16: According to your detailed instructions, we summarized key findings at the end (see more details in lines 837-849). For the second question, Table 1 of the article presents a description of transcriptome sequencing in Chinese sturgeon based on PacBio sequencing platform, which does not involve any significance comparison, so we did not add significance level of p<0.05 in table 1 of the revised manuscript.
Comments 17: Population Decline Details: Include specific statistics or studies?
Response 17: Sure. Due to negative factors such as overfishing and habitat degradation, the populations of Chinese sturgeon have been severely depleted in the past thirty years, with a sharp reduction in distribution areas. At present, Chinese sturgeon only appears in the Yangtze River, with the only known spawning ground located within 4 km downstream of the Gezhouba Dam; no traces of the Chinese sturgeon have been found in other rivers of its natural distribution (Zhu et al., 2024, and Wei et al., 2020). We have incorporated specific statistics from the referenced studies in the paper to enhance the content. Please see more details in the lines 61-65 of the revised manuscript.
Comments 18: Discuss any successes or challenges faced in artificial propagation efforts? How understanding gene expression patterns can directly contribute to conservation strategies and management practices?
Response 18: Thanks for your nice advice. Artificial propagation of Chinese sturgeon has indeed encountered several challenges, such as issues related to diseases, a prolonged rearing cycle, and the potential impacts of environmental pollution. I would like to address this from two perspectives:
Improving Habitat Management: Insights gained from tissue-specific gene expression can deepen our understanding of the ecological requirements of Chinese sturgeon. This information is vital for effective habitat restoration and management efforts, ensuring that conservation strategies align with the species' biological needs.
Informing Policy and Advocacy: The findings from our current study can provide scientific evidence to bolster conservation policies and initiatives. By emphasizing the ecological and evolutionary significance of Chinese sturgeon, we can advocate for stronger protections and additional resources for its conservation.
We have added this content in the discussion section. Please see more details in lines 826-835 of the revised manuscript.
Comments 19: Discuss the implications of the variation in tissue-specific genes, particularly why brain tissue has a higher number compared to others. Expand on the significance of tissue-specific gene expression by discussing its implications for understanding physiological adaptations to environmental changes?
Response 19: Sure. The variation in tissue-specific genes is a crucial area of study, especially when considering the higher number identified in brain of the Chinese sturgeon (Acipenser sinensis) compared to other tissues. This phenomenon has several implications, such as (1) neurological complexity: The brain plays a vital role in integrating environmental signals and coordinating responses. A higher number of tissue-specific genes in this organ suggests increased complexity and specialization, which may be necessary for processing sensory information, regulating behavior, and facilitating complex physiological functions. This could indicate that the brain of Chinese sturgeon is particularly adapted to its ecological niche, allowing for improved survival and reproduction; (2) physiological Responses: Tissue-specific gene expression is fundamental to the physiological adaptations organisms make in response to environmental changes. In the case of Chinese sturgeon, the higher gene expression in brain may facilitate rapid and effective responses to environmental challenges. This adaptability could be a key factor in the species' resilience to factors such as climate change and habitat loss. In summary, the higher number of tissue-specific genes in the brain of Chinese sturgeon not only reflects the complexity of its neurological processes but also underscores the significance of gene expression in facilitating physiological adaptations to environmental changes.
We have added this content in the discussion section of the revised manuscript. Please see more details in lines 708-722 of the revised manuscript.
Comments 20: How the comparisons enhance our understanding of evolutionary biology in sturgeons?
Response 20: We apologize for any confusion caused by our previous description. In fact, our present study primarily investigates the specificity of gene expression in different tissues of Chinese sturgeon. This research lays a foundation for understanding the fate of various genes following the whole-genome duplication in Chinese sturgeon, thereby enhancing our knowledge of the evolution and expression of duplicated genes. Please find related changes in lines 813-816.
Comments 21: Start with a concise summary of the most significant findings before the details.?
Response 21: Thank you for your suggestion. We recognize that our writing contained some redundancy, and in this version, we've incorporated a more concise summary to clearly present our main findings. Please see more details in lines 22-28 of the revised manuscript.
Comments 22: Expand on the significance of the identified functional pathways and biological processes associated with housekeeping genes, perhaps linking them to broader physiological contexts.?
Response 22: Thank you for your advice. Yes, we have addressed your recommendation by expanding the discussion on housekeeping genes. Please see more details in lines 738-753 of the revised manuscript.
Comments 23: Emphasize the importance of making your transcriptome dataset publicly available? End with a strong concluding statement that summarizes the overall significance of your work and its potential impact on future studies related to aquaculture and conservation biology.?
Response 23: Thank you for your valuable advice. We have revised the manuscript to emphasize the importance of making our transcriptome dataset publicly available. Additionally, we have included a strong concluding statement that summarizes the overall significance of our work and its potential impact on future studies in aquaculture and conservation biology. Please see more details in lines 671-677, and lines 817-835 of the revised manuscript.

Reviewer 4 Report
Comments and Suggestions for Authors
This manuscript presents a comprehensive gene expression analysis in the Chinese sturgeon (Acipenser sinensis), an ancient and complex autooctoploid species facing conservation challenges. The authors employ a thorough methodological approach, utilizing both PacBio Iso-seq and RNA-seq analyses across 11 tissues, generating a transcriptomic repository with over 25,000 full-length transcripts. The identification of housekeeping genes (HKGs) and tissue-specific genes (TSGs) offers valuable insights into their roles in various biological processes and tissue functions.
While the methodology employed is robust and exceeds what is typically required for the scope of this study, it does provide a solid foundation for future research, especially for reference gene selection in gene expression studies. The in-depth analysis of housekeeping genes and tissue-specific functions, particularly in processes like ribosome biogenesis and specific physiological pathways, adds significant value to the field.
However, the extensive depth of this study might be considered overly ambitious for addressing the primary research questions, which could have been answered with a more streamlined approach. Nevertheless, the data generated is highly valuable for future investigations into gene expression in Acipenser sinensis and related species. Overall, this work represents an important contribution to the field and will be a useful resource for future studies of gene expression and functional genomics in this species.
Minnor comments:
- In general, I find the manuscript to be overextended in its explanations. The necessary information should be provided, but without the excessive use of adjectives and lengthy, unnecessary details. The writing could be more concise and focused, avoiding redundant explanations.
- It is not clear if the experiments were approved by an “Animal Ethics Committee” or authors only follow Animal Ethics rules.
- is a DNase treatment used to avoid DNA contamination? How were the primers designed? Do they span an intron to ensure that the amplified product is the transcript and not genomic DNA?
Comments on the Quality of English Language
none
Author Response
Comments 1: In general, I find the manuscript to be overextended in its explanations. The necessary information should be provided, but without the excessive use of adjectives and lengthy, unnecessary details. The writing could be more concise and focused, avoiding redundant explanations.
Response 1: Thank you for your good advices. Yes, we have revised the entire manuscript to make it more focused and streamlined, eliminating redundant structures. We hope the revised version will meet your expectations.
Comments 2: It is not clear if the experiments were approved by an “Animal Ethics Committee” or authors only follow Animal Ethics rules.
Response 2: In fact, the experiments of this study were approved by the Animal Experiment Ethics Committee, Yangtze River Fisheries Research Institute, Chinese Academy of Fishery Sciences (License no. YFI2021CPL02). We have added this sentence in the revised manuscript (see lines 874-876).
Comments 3: Is a DNase treatment used to avoid DNA contamination? How were the primers designed? Do they span an intron to ensure that the amplified product is the transcript and not genomic DNA?
Response 3: Yes, DNase treatment is commonly used to eliminate DNA contamination in RNA samples prior to reverse transcription or PCR. This ensures that any presented DNA will be degraded, allowing for amplification of only the cDNA derived from mRNA.
Primer pairs are typically designed to span an intron to confirm that the amplified product is derived from target mRNA, thereby distinguishing it from genomic DNA. This means that the primers should anneal to sequences located on either side of an intron in the gene of interest, ensuring that if genomic DNA is present, the primer pairs will not produce an amplicon because they cannot span the intron.
Overall, these practices help to ensure high specificity and accuracy of the results when studying gene expression.

Reviewer 5 Report
Comments and Suggestions for Authors
Li et al. conducted a study titled 'Expression Profiles of Housekeeping Genes and Tissue-Specific Genes in Different Tissues of Chinese Sturgeon". Below are some comments and suggestions regarding their work.
- I suggest that the authors include the scientific name of the fish species in the title for clarity and precision.
- Lines 48-50, based on the authors' findings and data analysis, most results pertain to fundamental biological functions and are presented as general predictions. Therefore, I believe these findings are not sufficiently novel to be considered groundbreaking.
- The Methods and Materials section in the current study is well-organized. However, the presentation feels artificial, as though it were not written by a person. The writing style, structure, and content are overly generic, making it unsuitable to be considered a genuine research M&M section.
- As we know, the applications of omics approaches have become increasingly common and popular. Techniques such as RNA-seq or Iso-seq, along with others, often yield similar predictions using databases like GO, KEGG, NR, and SwissProt. However, I believe these data alone are insufficient to be considered novel. Furthermore, it remains unclear how they contribute meaningfully to understanding genetic and evolutionary aspects.
- In the discussion, the authors primarily describe their findings but fail to integrate them effectively. Additionally, the discussion does not address the research gaps and questions outlined by the authors (Lines 145-159). Although the study has broad aims, the discussion lacks alignment with those aims and does not demonstrate how the findings are connected.
- The references are outdated.
- Animal ethics approval should be provided in this study.
Author Response
Comments 1: I suggest that the authors include the scientific name of the fish species in the title for clarity and precision.
Response 1: Yes, it is done (see more details in line 4 of the revised manuscript).
Comments 2: Lines 48-50, based on the authors' findings and data analysis, most results pertain to fundamental biological functions and are presented as general predictions. Therefore, I believe these findings are not sufficiently novel to be considered groundbreaking.
Response 2: Thank you for your suggestion. We appreciate your feedback and have revised the main text to reduce the emphasis on certain speculations. Please see more details in the lines 49-51 of the revised manuscript.
Comments 3: The Methods and Materials section in the current study is well-organized. However, the presentation feels artificial, as though it were not written by a person. The writing style, structure, and content are overly generic, making it unsuitable to be considered a genuine research M&M section.
Response 3: Thank you for your feedback. As English is not our mother language, there may be some shortcomings in our expression. In this revised manuscript, we have made further improvements to meet your expectations, especially with assistance from a colleague who had worked in the USA for over nine years. Additionally, we also obtained English editing service from EditorBar (no. LE202411010015).
Comments 4: As we know, the applications of omics approaches have become increasingly common and popular. Techniques such as RNA-seq or Iso-seq, along with others, often yield similar predictions using databases like GO, KEGG, NR, and SwissProt. However, I believe these data alone are insufficient to be considered novel. Furthermore, it remains unclear how they contribute meaningfully to understanding genetic and evolutionary aspects.
Response 4: Thank you for your comments. You are right. It is indeed true that with advancements in sequencing technology, RNA-seq and Iso-seq have become more standardized. We have revised the wording and removed references to novelty, please see more details in line 167 of the revised manuscript. Despite the progress in sequencing techniques, certain species remain under-researched and represent gaps in knowledge. The Chinese sturgeon has garnered increasing attention in evolutionary and genetic studies in recent years. This interest may stem from various factors, such as its status as an ancient fish species. Although it belongs to the ray-finned fishes, its morphology and structure (cartilage) bear similarities to sharks, which are cartilaginous fish. Additionally, it is an octoploid species, yet its population is limited, leading to a high conservation status. Therefore, the data generated from our research holds significant importance for understanding the evolutionary characteristics of Chinese sturgeon.
Comments 5: In the discussion, the authors primarily describe their findings but fail to integrate them effectively. Additionally, the discussion does not address the research gaps and questions outlined by the authors (Lines 145-159). Although the study has broad aims, the discussion lacks alignment with those aims and does not demonstrate how the findings are connected.
Response 5: Thank you for your valuable feedback. We have made more in-depth revisions to the discussion section in line with your suggestions. See more details in lines 671-695; 708-722; 738-753, and 811-835 of our revised manuscript.
Comments 6: The references are outdated.
Response 6: Thanks for your instructive advice. We have added some recent references such as nos. 1, 4, 53, 54, 56, 60, 61, 63 and 64. The changed references were highlighted in blue of our revised version.
Comments 7: Animal ethics approval should be provided in this study.
Response 7: Thanks for your nice advice. Yes, it is done (lines 874-876).

Round 2
Reviewer 1 Report
Comments and Suggestions for Authors
The paper can be accepted in the present form